# Dywave: Event-Aligned Dynamic Tokenization for Heterogeneous IoT Sensing Signals

Tomoyoshi Kimura [1]  Denizhan Kara [1]  Jinyang Li [1*]  Hongjue Zhao [1]  Yigong Hu [1]  Yizhuo Chen [1]
Xiaomin Ouyang [2]  Shengzhong Liu [3]  Tarek Abdelzaher [1]

## Abstract

Internet of Things (IoT) systems continuously collect heterogeneous sensing signals from ubiquitous sensors to support intelligent applications such as human activity analysis, emotion monitoring, and environmental perception. These signals are inherently non-stationary and multi-scale, posing unique challenges for standard tokenization techniques. This paper proposes Dywave, a dynamic tokenization framework for IoT sensing signals that constructs compact input representations aligned with intrinsic temporal structures and underlying physical events. Dywave leverages wavelet-based hierarchical decomposition, identifies meaningful temporal boundaries corresponding to underlying semantic events, and adaptively compresses redundant intervals while preserving temporal coherence. Extensive evaluations on five real-world IoT sensing datasets across activity recognition, stress assessment, and nearby object detection demonstrate that Dywave outperforms state-of-the-art methods by up to 12% in accuracy, while improving computational efficiency by reducing input token lengths by up to 75% across mainstream sequence models. Moreover, Dywave exhibits improved robustness to domain shifts and varying sequence lengths.

## 1. Introduction

Internet of Things (IoT) systems increasingly rely on continuous streams of sensing modalities, such as inertial measurement unit (IMU) for human activity recognition (HAR) (Korany et al., 2019; Kawano et al., 2023; Wang et al., 2022), electrocardiograms (ECG) for healthcare (Nath et al., 2023; Alharbi et al., 2023; Zakaria et al., 2023), and acoustic signals that increase environmental awareness to enhance human safety (Wang et al., 2023b; Kim et al., 2023; Kimura et al., 2024). Learning from these diverse signals enables intelligent sensing applications that can perceive, understand, and respond to the physical world (Baris et al., 2025).

Recent advances in language and vision highlight the central role of *data tokenization* in enabling large-scale generalizable models (Petrov et al., 2023; Bommasani et al., 2021). In NLP, words and subwords form linguistically and statistically grounded discrete tokens (Sennrich et al., 2016; Kudo & Richardson, 2018), whereas in vision, spatial patches serve as localized tokens aligned with model inductive biases (He et al., 2016; Dosovitskiy et al., 2020). These tokenization schemes establish a shared representation interface for large-scale training, zero-shot transfer, and task generalization (Ahia et al.; Tao et al., 2024). In contrast, raw signals in IoT sensing applications are often non-intuitive for humans and lack a similar natural notion of semantic units. Unlike words or images, signals appear as continuous waveforms that are temporally heterogeneous, with information encoded in the transitions between underlying physical events and in multiscale interactions, such as fast transients overlapping with slow contextual trends. The absence of appropriate atomic units for sensing signals creates a *tokenization gap*, forcing existing approaches to rely on uniform windows that partition signals into *patches* as input tokens for downstream backbones (Nie et al., 2023; Naghashi et al., 2025; Ekambaram et al., 2023). Here, we consider *uniform patching* as a method that partitions signals into fixed-size patches, either at a single temporal scale (Nie et al., 2023) or across multiple resolutions (Naghashi et al., 2025; Wang et al., 2024). Since these windows are defined a priori, their boundaries are inherently content-agnostic and often misaligned with the underlying signal dynamics.

**Limitations**: Although tokens corresponding to uniform windows offer a simple heuristic approach, it remains misaligned with the intrinsic dynamic structures of physical events, which rarely conform to uniform timescales. This results in event fragmentation and the obscuration of underlying semantics. For example, in human activity recognition

[1] University of Illinois Urbana-Champaign [2] Hong Kong University of Science and Technology [3] Shanghai Jiao Tong University. *Correspondence to: Jinyang Li <jinyang7@illinois.edu>.

*Proceedings of the 43rd International Conference on Machine Learning*, Seoul, South Korea. PMLR 306, 2026. Copyright 2026 by the author(s).

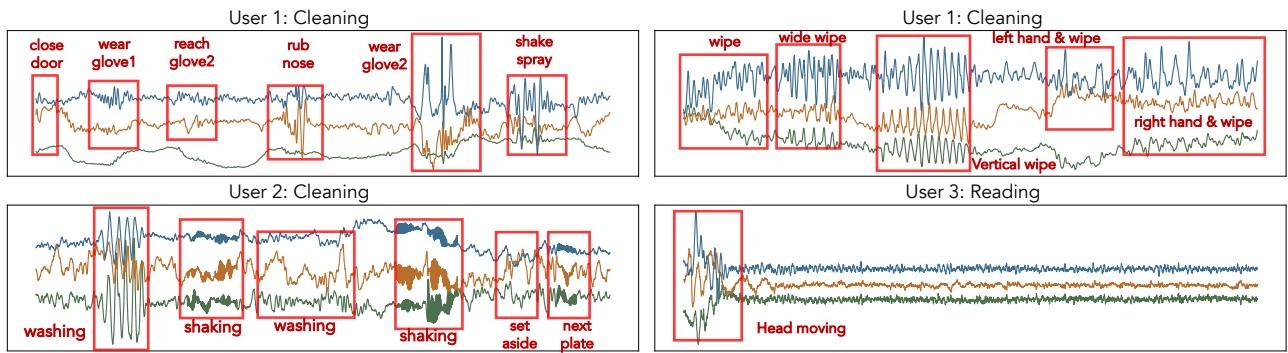

*Figure 1.* Ego4D (HAR) raw signal examples. Signal events are manually annotated with red bounding boxes.

using IMU signals, brief motion gestures (*e.g.*, waving) may occur within a second, while complex activities (*e.g.*, walking) can span tens of seconds and vary in intensity.

Moreover, real-world signals exhibit highly irregular information density, with quiescent intervals alternating with short bursts of salient activity. Uniform patching can generate abundant patches over long periods of redundancy that contribute little information, resulting in equal computation being allocated to both informative and noninformative regions, dramatic inflation of input length, and limited representativeness of critical transitions.

Lastly, optimal hyperparameters (*e.g.*, patch size and stride) are application-specific. Smaller strides preserve fine-grained details but explode sequence length and computation cost, whereas larger non-overlapping patches can compress computation at the expense of semantic precision. Performance fluctuates irregularly across these settings, revealing a lack of universally effective configurations and necessitating time-consuming per-domain tuning.

**Methodology.** We argue that effective modeling of sensing signals requires rethinking *input tokenization* as a dynamic process rather than a preset, fixed heuristic. Instead of uniformly segmenting time-series and modifying the backbone architecture, we propose Dywave, a *dynamic, event-aligned tokenization* scheme that adapts to the intrinsic temporal structure of physical signals and is compatible with mainstream backbone encoders.

To achieve this, Dywave first extracts **hierarchical embeddings** by explicitly leveraging the scale-separated structure of physical events. These structures enable Dywave to exploit multi-resolution temporal patterns, rather than treating the signal as a homogeneous sequence.

Building on these embeddings, Dywave performs **temporal anchor formation** by estimating which timesteps are most salient and selecting anchors at semantic transitions that correspond to meaningful events.

Finally, Dywave applies **dynamic temporal fusion**, aggregating neighboring timesteps into anchor-aligned tokens

through saliency-weighted pooling. This produces compact representations whose length adapts to semantic complexity rather than raw signal duration.

**Evaluation**: We evaluate Dywave on five diverse real-world sensing datasets spanning diverse sampling rates, sequence lengths, and temporal dynamics. Dywave mitigates fragmentation and truncation by focusing on semantically meaningful intervals. Furthermore, we demonstrate how Dywave decomposes complex human activities into fine-grained micro-activity segments, revealing the underlying temporal structures of human-centric continuous sensor signals.

The contributions are summarized as follows:

- We identify the *tokenization gap* in physical sensing, highlighting the lack of human-intuitive semantic units.

- We propose Dywave, a dynamic tokenization module that converts raw signals into compact, event-aligned tokens.

- Extensive evaluation on real-world applications with a case study demonstrates Dywave's superior downstream performance and fine-grained event decomposition.

## 2. Motivation and Design Principles

This section outlines challenges in sensing signal tokenization and introduces the design principles behind Dywave.

### 2.1. Challenges and Motivations

Unlike words or objects in language and vision, sensing signals appear as continuous waveforms where event semantics are dispersed across time and encoded in transitions. Existing methods segment these streams into predefined windows uniformly across time. However, these segments rarely correspond to coherent physical events or human actions, as temporal dynamics are often irregular and context-dependent, introducing unique challenges.

**Signal Heterogeneity and Complexity**. Real-world sensing data are extremely diverse across users, contexts, and modalities. Even when performing the same activity, dif-

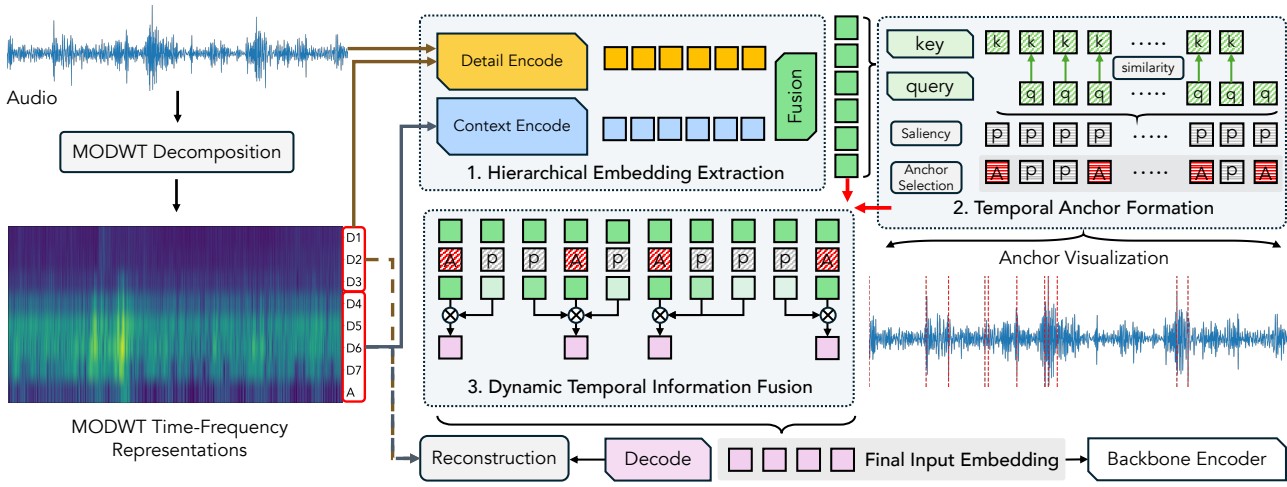

*Figure 2.* Overview of Dywave.

ferent users produce signals that vary greatly in temporal structure and intensity. To illustrate this variability, Figure 1 visualizes 30-second accelerometer samples from the Ego4D human activity recognition dataset (Grauman et al., 2022), comparing signals of *cleaning* activity across users and time periods, as well as the *reading* activity. Even within the same activity, signal patterns differ significantly, both across users and across sessions of the same user. For instance, User 1's two cleaning segments show distinct motion rhythms, while User 2's cleaning activity exhibits sharper, more intense movements. Conversely, User 3's reading activity is dominated by long stationary intervals with only brief bursts of motion at the beginning. The variations highlight the non-stationary, user-dependent nature of sensing data, where activity semantics are tightly coupled with individual context. Applying uniform patching under such diversity neglects signal-specific semantics, producing arbitrary segmentations that fail to align with true event boundaries or preserve coherence across similar activities.

**Computation Efficiency**. Beyond representational granularity, a uniform window for patching also limits computational efficiency. In ubiquitous computing systems with strict latency and energy constraints, uniform patching treats all regions equally, allocating identical computation to both dynamic and redundant segments. As illustrated in Figure 1, long and flat intervals of User 3's reading activity are dominant. Yet, uniform tokenization can expand the interval with minimal semantic content, unnecessarily inflating the input length and the computational cost.

**2.2. Design Principles for Sensing Signal Tokenization**

The challenges above reveal the limitations of uniform patching for heterogeneous, non-stationary sensing signals and underscore the need for a dynamic tokenization that moves beyond static, uniform windows. From these observations, we derive two design principles for effectively tokenizing

time-series sensing signals.

**Principle 1: Physical grounding.** Sensing signals originate from continuous physical processes, where semantics emerge from transitions between underlying physical states. Therefore, tokenization must preserve physical coherence to ensure each token corresponds to a distinct, meaningful physical event rather than an arbitrary temporal slice. Physically grounded representations should maintain the temporal continuity of event relationships to infer contextually consistent semantics for dynamic sensing signals.

**Principle 2: Adaptivity across scales and domains.** Sensing signals are multiscale and exhibit strong temporal heterogeneity, with rapid transient spikes coexisting with long, gradual dynamics. Hence, an effective tokenization strategy should balance temporal and semantic resolution by allocating finer granularity to information-dense events and coarser granularity to stable segments. Moreover, the tokenization strategy should adapt on a per-sample basis, rather than assuming homogeneous temporal structure across segments.

By adhering to these principles, a dynamic tokenization for sensing signals can transform raw sensor streams into structured, semantically aligned representations, enabling more robust and efficient downstream learning.

## 3. Dywave Design

This section presents Dywave, a dynamic tokenization module that adapts to the signal's underlying temporal structure. We provide a detailed overview of Dywave in Figure 2.

**Problem Formulation**: Let $X \in \mathbb{R}^{C \times L}$ denote a raw time-series segment, where $C$ is the number of channels and $L$ is the sequence length. The goal of Dywave is to transform $X$ into a compact sequence of *patched tokens* $E \in \mathbb{R}^{C \times L' \times d}$, where $L'$ depends on the sample semantics.

## 3.1. Hierarchical Embedding

Real-world time-series signals exhibit multi-granular structure across temporal and frequency scales. To capture this, we apply the *Maximal Overlap Discrete Wavelet Transform* (MODWT) (Percival & Walden, 2000) to decompose raw inputs into multi-resolution time-frequency representations. For an input $X \in \mathbb{R}^{C \times L}$, MODWT yields:

$$\{dX_1, \ldots, dX_J, A\} = \text{MODWT}(X), \qquad (1)$$

where $A \in \mathbb{R}^{C \times L}$ encodes long-term global trends, $dX_1 \in \mathbb{R}^{C \times L}$ captures highest-frequency variations, and $\{dX_j\}_{j \geq 2}$ represent progressively slower oscillations. Since MODWT is undecimated, all components preserve the original sequence length $L$.

Next, we partition the components into detail and context streams and extract hierarchical embeddings that capture the fine-grained transients and long-range temporal structure.

**Detail Embedding.** The detail stream captures localized, high-frequency variations. We project it using lightweight convolutional layers, which model short-range dependencies while preserving temporal alignment.

$$E^U = \text{DetailEncoder}(\{X, dX_1, \ldots, dX_K\}). \qquad (2)$$

**Context Embedding.** The context embedding encodes slow-varying, long-range patterns through a lightweight hourglass transformer as the context encoder, which performs downscaling, self-attention, and upscaling. The downsampling degree is adaptively selected to balance model capacity and computational cost.

$$E^V = \text{ContextEncoder}(\{dX_{K+1}, \ldots, dX_J, A\}). \qquad (3)$$

**Embedding Fusion.** We fuse detail and context embeddings along the feature dimension to form a unified hierarchical embedding $E^F = \text{Concat}(E^U, E^V)$ that is temporally aligned and semantically enriched for downstream tasks.

## 3.2. Temporal Anchor Formation

Rather than treating segmentation as an explicit objective, Dywave adaptively allocates temporal resolution based on intrinsic signal dynamics. Given hierarchical embeddings that jointly encode fine-grained transients and long-range context, Dywave identifies regions where representational detail should be preserved at higher density, while compressing temporally redundant intervals. As a result, patch boundaries emerge implicitly from variations in underlying physical dynamics, instead of being imposed by uniform temporal intervals or predefined heuristics.

**Temporal Event Saliency Estimation.** To quantify where finer temporal resolution is warranted, Dywave estimates event saliency by measuring representational change between adjacent timesteps.

$$P_t = 1 - \text{sim}(F_k(E^F_{t-1}), F_q(E^F_t)), \qquad t \in [2, L], \quad (4)$$

where $\text{sim}(\cdot)$ denotes cosine similarity and $F_k, F_q$ are linear layers that map $E^F$ to key and query.

Intuitively, continuous physical processes tend to produce slowly varying embeddings, resulting in low saliency scores, whereas genuine event transitions induce abrupt representational shifts that yield high saliency.

**Anchor Allocation.** $P_t$ acts as a continuous indicator of local information density that guides subsequent resolution allocation. However, $P_t$ may exhibit local fluctuations due to noise or minor signal variations. To regulate density and avoid over-allocation, Dywave applies temporal non-maximum suppression and top-$k$ selection to extract a compact set of anchors.

$$\mathcal{A} = \text{TopK}(\text{NMS}(P, w_{\text{nms}}), \lceil \tau \cdot L \rceil), \qquad (5)$$

where $w_{\text{nms}} = \lfloor L/(2\lceil \tau L \rceil) \rfloor$ is the window size for NMS, and $\lceil \tau \cdot L \rceil$ is the maximum number of anchors to select. The resulting anchors $\mathcal{A}$ indicate where representational capacity should be concentrated, enabling adaptive compression that reflects semantic complexity instead of raw signal duration.

## 3.3. Dynamic Temporal Information Fusion

Following temporal anchor formation, Dywave performs resolution-aware temporal fusion to construct compact input representations that reflect semantic complexity. Fusion is guided by anchor locations $\mathcal{A}$ that indicate where higher representational fidelity is required, allowing Dywave to concentrate computation around salient physical events while smoothly compressing temporally redundant intervals commonly observed in real-world sensing signals.

**Temporal Anchor Aggregation.** Temporal anchors act as sparse reference points around which local temporal neighborhoods are aggregated. Rather than enforcing hard partitions, each timestep is associated with its nearest anchor based on temporal proximity. Formally, the anchor assignment for time $t$ is defined as:

$$\kappa(t) = \arg \min_{a \in \mathcal{A}} |t - a|, \qquad t \in \{1, \ldots, L\}. \qquad (6)$$

**Saliency-Weighted Temporal Fusion.** Given anchor-centered temporal neighborhoods, Dywave aggregates embeddings using saliency-aware weighting. Timesteps exhibiting greater information variability contribute more prominently to the fused representation, while stable regions are down-weighted. For each anchor $a_k \in \mathcal{A}$, the final embedding $E = \{E_1, \ldots, E_{|\mathcal{A}|}\}$ is computed as:

$$E_k = \frac{\sum_{t: \kappa(t) = a_k} P_t E^F_t}{\sum_{t: \kappa(t) = a_k} P_t + \varepsilon}, \qquad k \in 1, \ldots, |\mathcal{A}|; \qquad (7)$$

*Table 1.* Short-context classification performance. Best performance is **bolded** and second best is underlined.

| Model | Dataset | MOD seismic | | PAMAP2 acc | | RWHAR acc | | PAMAP2 gyro | | RWHAR gyro | |
|---|---|---|---|---|---|---|---|---|---|---|---|
| | | Acc | F1 | Acc | F1 | Acc | F1 | Acc | F1 | Acc | F1 |
| **Transformer** | PatchTST (Nie et al., 2023) | *0.7743* | *0.7653* | 0.7629 | 0.7549 | *0.8840* | *0.8871* | *0.6498* | *0.6343* | *0.6740* | *0.5804* |
| | MedFormer (Wang et al., 2024) | 0.4059 | 0.3581 | *0.7709* | *0.7792* | 0.8240 | 0.8160 | 0.6317 | 0.6111 | 0.5239 | 0.4308 |
| | DropPatch (Qiu et al., 2025) | 0.7582 | 0.7455 | 0.6565 | 0.6432 | 0.6809 | 0.6414 | 0.3876 | 0.3256 | 0.4980 | 0.3764 |
| | WaveToken (Masserano et al., 2025) | 0.5358 | 0.5136 | 0.4381 | 0.3658 | 0.2908 | 0.2128 | 0.1475 | 0.0624 | 0.2591 | 0.1595 |
| | Multi-Patch (Naghashi et al., 2025) | 0.6739 | 0.6598 | 0.7512 | 0.7388 | 0.8627 | 0.8357 | 0.6037 | 0.5803 | 0.5609 | 0.4701 |
| | **Dywave (ours)** | **0.7917** | **0.7922** | **0.7977** | **0.7905** | **0.9094** | **0.8932** | **0.7118** | **0.6948** | **0.7294** | **0.7407** |
| **Mamba2** | PatchTST (Nie et al., 2023) | *0.7696* | *0.7582* | *0.7488* | *0.7445* | *0.7340* | *0.7429* | 0.7058 | 0.6881 | *0.7744* | *0.7758* |
| | MedFormer (Wang et al., 2024) | 0.7006 | 0.6873 | 0.7019 | 0.6770 | 0.7144 | 0.6267 | 0.7019 | 0.6770 | 0.7236 | 0.7140 |
| | DropPatch (Qiu et al., 2025) | 0.7525 | 0.7450 | 0.5110 | 0.4824 | 0.4126 | 0.2915 | 0.3644 | 0.3302 | 0.4126 | 0.2915 |
| | WaveToken (Masserano et al., 2025) | 0.4233 | 0.3867 | 0.5919 | 0.5569 | 0.3589 | 0.2889 | 0.3454 | 0.2638 | 0.4951 | 0.4027 |
| | Multi-Patch (Naghashi et al., 2025) | 0.7033 | 0.6993 | 0.7307 | 0.7176 | 0.7224 | 0.5944 | *0.7307* | *0.7176* | 0.7380 | 0.7259 |
| | **Dywave (ours)** | **0.8078** | **0.8010** | **0.8072** | **0.7980** | **0.8517** | **0.8225** | **0.7338** | **0.7272** | **0.7761** | **0.7959** |

where $\varepsilon$ ensures numerical stability, and $E \in \mathbb{R}^{C \times |\mathcal{A}| \times d}$ is the final embedding for the backbone encoder.

The saliency-weighted fusion prioritizes temporally localized event dynamics while suppressing redundant background activity, enabling adaptive compression that preserves semantic integrity without explicit segmentation.

### 3.4. Training Objective

To ensure $E$ preserves multi-scale information, Dywave leverages a lightweight decoder to reconstruct the wavelet coefficients from the fused embedding $E$:

$$\mathcal{L}_{\text{rec}} = \text{MSE}\big(\{dX_1, \ldots, dX_J, A\}, \text{Decode}(E)\big), \quad (8)$$

where Decode comprises a linear layer, transposed convolution, and adaptive pooling to match input length $L$. This enforces that compressed tokens retain both low-frequency trends and high-frequency transients. The full objective combines training supervision with reconstruction:

$$\mathcal{L} = \mathcal{L}_{\text{task}} + \lambda_{\text{rec}} \cdot \mathcal{L}_{\text{rec}}. \quad (9)$$

Due to space limitations, we provide additional details on Dywave's components in Appendix C.

## 4. Evaluation

We evaluate Dywave on five sensing datasets, assessing short/long-context, multimodal, and generalization performance, as well as token efficiency. We also conduct ablations and qualitative analyses of Dywave's tokenization.

### 4.1. Evaluation Setup

**Datasets.** We evaluate on 5 datasets across three sensing applications: Human Activity Recognition (Grauman et al., 2022; Reiss & Stricker, 2012; Sztyler & Stuckenschmidt, 2016), Moving Object Detection (Liu et al., 2023), and Stress Assessment (Schmidt et al., 2018b). Each sample

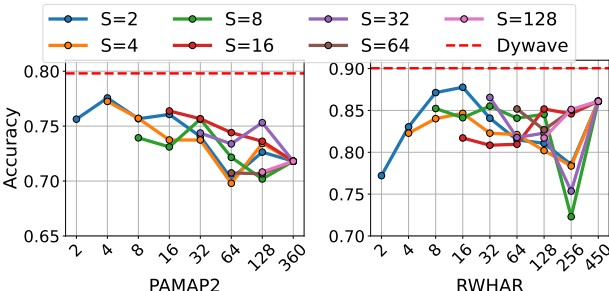

*Figure 3.* Short-context performance vs. different parameters.

is a fixed-length time-series segment used for short- and long-context classification.

**Baselines.** We consider 5 baselines compatible with various backbones: PatchTST (Nie et al., 2023), DropPatch (Qiu et al., 2025), MedFormer (Wang et al., 2024), WaveToken (Masserano et al., 2025), and MultiPatch (Naghashi et al., 2025). We evaluate them using two sequence encoders, Transformer (Vaswani et al., 2017) and Mamba2 (Dao & Gu, 2024), with the same parameter settings.

We provide more details on the datasets, baselines, encoders, and additional configurations in Appendix A, B, and D.

### 4.2. Short-context Performance

#### 4.2.1. SHORT-CONTEXT CLASSIFICATION

We evaluate Dywave on short-context signals (2-5 seconds) across five sensing configurations: MOD seismic, PAMAP2 accelerometer & gyroscope, and RWHAR accelerometer & gyroscope. Table 1 shows classification performance using Transformer and Mamba2 backbone encoders. Dywave consistently achieves the best performance across all datasets and architectures, with gains up to 12% in accuracy on HAR tasks using Mamba2. Unlike fixed-size tokenization methods, Dywave eliminates the need for exhaustive hyperparameter tuning of patch size and stride. As shown in Figure 3, PatchTST is highly sensitive to these param-

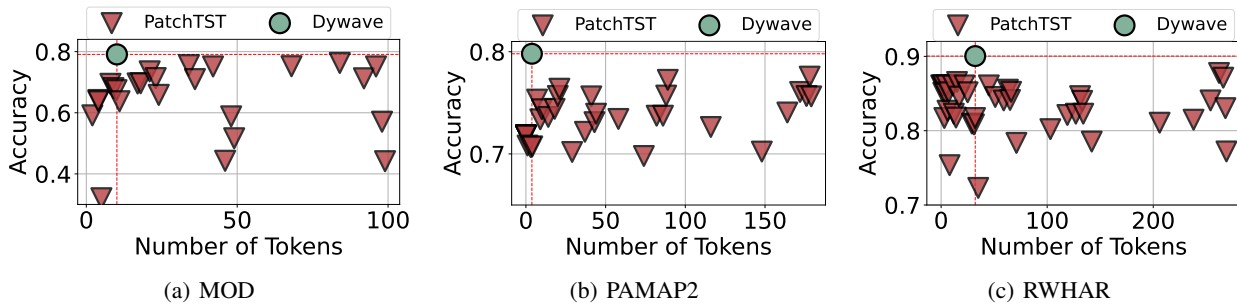

(a) MOD        (b) PAMAP2        (c) RWHAR

*Figure 4.* Short-context Accuracy vs. Token with the Transformer encoder.

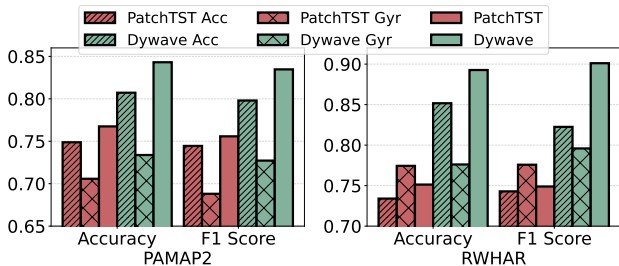

*Figure 5.* Multimodal classification accuracy.

| Model | Datasets | RWHAR | | | PAMAP2 | | |
|---|---|---|---|---|---|---|---|
| | | Acc | F1 | # Tokens | Acc | F1 | # Tokens |
| **T** | LightGTS | 0.8586 | 0.8513 | **14.66** | 0.6830 | 0.6621 | *10.71* |
| | Ruptures | 0.7761 | 0.7302 | 57.60 | 0.6877 | 0.6604 | 37.85 |
| | Dywave | **0.9094** | **0.8932** | *29.61* | **0.7977** | **0.7905** | **3.76** |
| **M** | LightGTS | 0.7917 | 0.7409 | **14.66** | 0.7003 | 0.6977 | *10.71* |
| | Ruptures | 0.7998 | 0.7665 | 57.60 | 0.6629 | 0.6341 | 37.85 |
| | Dywave | **0.8517** | **0.8225** | *34.65* | **0.8072** | **0.7980** | **2.23** |

*Table 2.* Comparing Dywave with dynamic patching baselines. T stands for Transformer and M stands for Mamba2.

eters, requiring extensive grid search, while Dywave uses learnable, instance-specific segmentation.

Moreover, Wavetoken performs notably worse than other baselines. Discretizing the input into quantized token IDs appears ill-suited for high-frequency sensing data with rich dynamics, as it disrupts the fine-grained amplitude and temporal coherence essential for signal characterization.

### 4.2.2. MULTIMODAL CLASSIFICATION

In addition to unimodal classification, we further validate Dywave's capability in handling multimodal sensing inputs by jointly using accelerometer and gyroscope signals. Each modality is processed independently by Dywave, producing modality-specific token sequences that are fed into separate backbone encoders. We then perform late fusion of intermediate representations before the final classifier layers. Figure 5 presents the classification results on PAMAP2 and RWHAR using the Mamba2 backbone encoder. Although Dywave's adaptive tokenization is not specifically designed to handle multimodal inputs, it still maintains strong performance across all datasets. Additionally, fixed-size tokenization requires extensive hyperparameter tuning to ensure consistent temporal resolution and alignment between modalities. The results demonstrate that Dywave generalizes well to multimodal input and remains robust and effective on adaptive segmentation.

### 4.2.3. PERFORMANCE VS. TOKEN DISTRIBUTION

Figure 4 compares classification accuracy against average token count for Dywave and PatchTST across MOD, PAMAP2, and RWHAR. PatchTST shows no clear correla-

tion between token count and accuracy, with more tokens often performing worse due to fixed tokenization's sensitivity to hyperparameters. Dywave consistently achieves comparable or superior accuracy with far fewer tokens (upper-left region), demonstrating that adaptive tokenization captures informative segments with many fewer tokens.

### 4.2.4. DYNAMIC PATCHING BASELINE COMPARISON

We compare Dywave against two dynamic patching baselines, LightGTS (Wang et al., 2025) and Ruptures (Truong et al., 2020). Table 2 reports results on RWHAR and PAMAP2 with both backbone encoders. Dywave consistently outperforms both baselines, confirming that representation-driven, event-aligned tokenization is more accurate and efficient than sequence-level adaptive patching or statistical signal segmentation. LightGTS applies uniform windows within each sample, which overlooks intra-sequence heterogeneity, where a single signal may contain both rapid transient events and stationary intervals. Ruptures performs change-point detection directly on the raw signal without task supervision, yielding boundaries statistically salient but not semantically meaningful, resulting in degraded performance while producing more tokens.

### 4.3. Long-context Performance

#### 4.3.1. LONG-CONTEXT CLASSIFICATION

We evaluate Dywave on long-context signals: MOD audio, Ego4D accelerometer, and WESAD ECG & EMG. Figure 6 shows results on both backbone encoders. Dywave consistently achieves the highest accuracy and F1-score across all datasets. The advantage is most significant

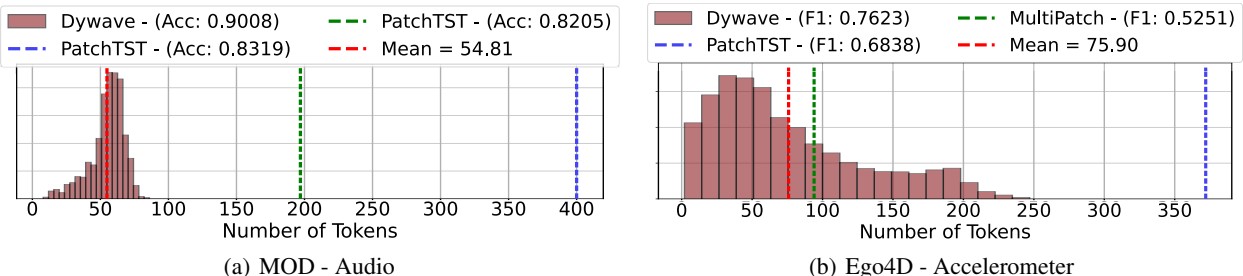

*Figure 6. Long-context* classification performance with the Transformer encoder.

*Figure 7.* Long-context token distribution with the Transformer encoder.

on Ego4D, where 30-second sequences contain multiple heterogeneous sub-events (posture transitions, hand-object interactions, environmental perturbations). Fixed-size tokenization mixes unrelated actions and obscures fine-grained transitions, while Dywave dynamically identifies semantic boundaries that align with activity transitions, enabling the backbone to focus on informative temporal regions.

### 4.3.2. Long-context Token Distribution

Figure 7 shows the distribution of input token lengths for Dywave compared to other methods. Dywave achieves higher accuracy and F1 scores while using significantly fewer tokens. On MOD audio (16,000 Hz), Dywave produces 55 tokens on average, nearly four times fewer than PatchTST, while maintaining superior accuracy. On Ego4D, Dywave delivers better performance with fewer tokens despite overlapping motion phases and ambient noise. These results demonstrate Dywave's ability to dynamically adjust tokenization to signal semantics, selectively retaining informative segments while compressing stationary intervals.

### 4.4. Generalizability Evaluation

We evaluate the generalization ability of Dywave under both *domain shifts* and *sequence-length variations*. Two finetuning strategies are considered: (1) *full backbone finetuning*, where the tokenization module is frozen and both the encoder and classification head are updated; and (2) *head-only finetuning*, where only the classification head is trained. We study (1) **cross-domain generalization** on out-of-domain MOD data, and (2) **sequence-length generalization** by transferring between Ego4D-S (5s) and Ego4D (30s).

### 4.4.1. Cross-Domain Generalization

As shown in Table 3, under full backbone finetuning, Dywave consistently improves cross-domain accuracy across both modalities while operating with substantially fewer tokens. In contrast, PatchTST shows limited robustness when the patching module is fixed.

This gap becomes more pronounced under head-only finetuning. In Table 4, PatchTST suffers severe performance degradation (*e.g.*, 0.81→0.34 on seismic), indicating strong reliance on source-domain embeddings. By comparison, Dywave maintains substantially higher accuracy, suggesting that its event-aligned tokenization encourages the backbone to learn more transferable representations during training rather than overfitting to domain-specific inputs.

### 4.4.2. Sequence-Length Generalization

On sequence-length generalization, Dywave exhibits stable performance when transferring between short and long temporal contexts, consistently outperforming PatchTST in both transfer directions with far fewer tokens. This indicates that Dywave can maintain the performance while representing longer temporal spans more efficiently. The advantage is more significant under head-only finetuning, where PatchTST significantly degrades in accuracy while Dywave preserves meaningful predictive performance. The token count gap further reveals Dywave's superior scalability. From 5s to 30 s, PatchTST increases tokens by over 6× (59→372), while Dywave grows modestly (15.5→35), demonstrating that token density is governed by semantic structure rather than signal length and Dywave can achieve length-invariant, low-latency input token representations with minimal redundancy.

*Table 3.* Generalization with **Full Backbone Finetuning**.

| Modality | MOD-Seismic | | | MOD-Audio | | | Ego4D-Accelerometer | | | Ego4D-Accelerometer | | |
|---|---|---|---|---|---|---|---|---|---|---|---|---|
| Type | Cross-Domain | | | Cross-Domain | | | Cross-Sequence: Short → Long | | | Cross-Sequence: Long → Short | | |
| Metrics | Acc | F1 | Num Tokens | Acc | F1 | Num Tokens | Acc | F1 | Num Tokens | Acc | F1 | Num Tokens |
| PatchTST | 0.8111 | 0.8091 | 43 | 0.7115 | 0.6999 | 400 | 0.6337 | 0.6718 | 372 | 0.5191 | 0.5001 | 59 |
| Dywave | **0.8119** | **0.8098** | **31.95** | **0.7485** | **0.7392** | **50.60** | **0.6400** | **0.6847** | **35.16** | **0.6212** | **0.6495** | **15.56** |

*Table 4.* Generalization with **Classification Head Finetuning**.

| Modality | MOD-Seismic | | | MOD-Audio | | | Ego4D-Accelerometer | | | Ego4D-Accelerometer | | |
|---|---|---|---|---|---|---|---|---|---|---|---|---|
| Type | Cross-Domain | | | Cross-Domain | | | Cross-Sequence: Short → Long | | | Cross-Sequence: Long → Short | | |
| Metrics | Acc | F1 | Num Tokens | Acc | F1 | Num Tokens | Acc | F1 | Num Tokens | Acc | F1 | Num Tokens |
| PatchTST | 0.3414 | 0.1273 | 43 | 0.4933 | 0.4663 | 400 | 0.3240 | 0.1905 | 372 | 0.3323 | 0.1819 | 59 |
| Dywave | **0.7565** | **0.7545** | **31.95** | **0.6873** | **0.6764** | **50.60** | **0.5326** | **0.5229** | **35.16** | **0.5545** | **0.5457** | **15.56** |

*Table 5.* Dywave variants with Transformer backbone encoder.

| Dataset | MOD seismic | | | MOD audio | | |
|---|---|---|---|---|---|---|
| | Acc | F1 | # Tokens | Acc | F1 | # Tokens |
| w/oWave | 0.7736 | 0.7673 | 80.39 | 0.8520 | 0.8474 | 174.50 |
| FixedDWT | 0.7669 | 0.7625 | 47.00 | 0.8399 | 0.8315 | 197.00 |
| w/oRecon | 0.7575 | 0.7493 | 56.69 | 0.7823 | 0.7831 | 32.95 |
| w/oFusion | *0.7863* | *0.7786* | 56.73 | 0.7582 | 0.7584 | *32.86* |
| CNNBound | 0.7676 | 0.7615 | 42.02 | 0.8064 | 0.8059 | 49.99 |
| SpecBound | 0.7683 | 0.7612 | *25.12* | *0.8533* | *0.8509* | 10.18 |
| Dywave | **0.7930** | **0.7861** | **16.90** | **0.9002** | **0.9001** | 50.34 |

## 4.5. Ablation Studies

We analyze contributions of different modules using five variants that remove individual components or replace designs. Table 5 shows results on MOD dataset for short- and long-context settings. Due to space limitations, we leave the detailed descriptions of the variants in Appendix E.1.

Variants **w/oWave**, **FixedDWT**, and **w/oRecon** evaluate the hierarchical embedding module. Removing or simplifying this module consistently degrades performance. Compared to Dywave-FixedDWT using standard DWT representations only, Dywave efficiently reduces the number of tokens and achieves superior accuracy. Dywave-w/oWave without hierarchical embedding fails to effectively reduce input tokens and can incur additional overhead. Removing the reconstruction (Dywave-w/oRecon) results in noticeable performance drop, as it encourages semantically coherent segmentation and retains fine-grained information. These results demonstrate that hierarchical embedding and reconstruction objective are critical for both efficacy and efficiency.

Variants **w/oFusion**, **CNNBound**, and **SpecBound** analyze dynamic anchor selection and fusion. Replacing saliency estimation with CNN-based prediction leads to a clear performance decline, indicating local convolutional filters are insufficient for capturing temporal dependencies. The event saliency module leverages contextual relations between neighboring segments, enabling semantically consistent segmentation. Dropping non-anchor segments (Dywave-w/oFusion) achieves comparable performance in short-

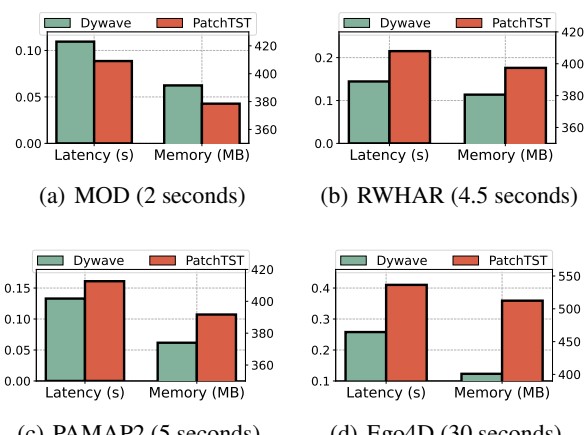

(a) MOD (2 seconds)     (b) RWHAR (4.5 seconds)

(c) PAMAP2 (5 seconds)     (d) Ego4D (30 seconds)

*Figure 8.* On-device (Raspberry Pi 4) Profiling.

context settings but sacrifices efficiency with a much higher token count. In long-context settings (MOD audio), it reduces input length but suffers greater accuracy degradation. This implies non-anchor segments contain meaningful cues and should not be discarded. Dynamic fusion is crucial for achieving compact representations without sacrificing accuracy. Using spectral energy as the saliency criterion captures low-level frequency changes in the signal but may not align with meaningful semantic transitions. Dywave-SpecBound occasionally produces lower token counts, but at the cost of reduced accuracy, further confirming that semantic-space saliency is more informative for event-aligned tokenization than signal-level spectral measures.

## 4.6. Computational Overhead

Figure 8 shows end-to-end runtime profiling on a Raspberry Pi 4 device. Dywave introduces preprocessing overhead relative to simple patching, but this cost is *amortized* by substantial reductions in backbone computation. For seismic, the signal is short and information-dense, leaving limited redundancy to compress. However, as the context length grows (*e.g.*, RWHAR, PAMAP2, Ego4D), compression reduces both latency and memory usage. For longer sequences

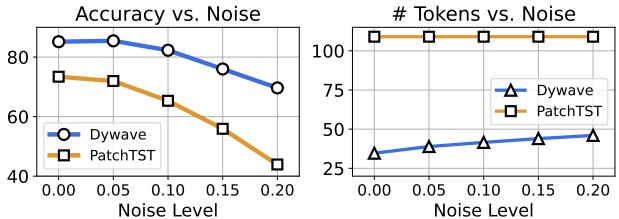

*Figure 9.* Inference robustness with random noise injection.

with heterogeneous dynamics, Dywave's token compression substantially reduces backbone computation, and the advantage grows with longer context windows or larger encoder models. This makes Dywave particularly well-suited for real-world deployments where signals are long, heterogeneous, and resource constraints are tight.

### 4.7. Inference Noise Robustness

Figure 9 evaluates robustness to Gaussian noise injected at test time on RWHAR with the Mamba2 backbone. PatchTST degrades sharply with increasing noise. On the other hand, Dywave degrades more gradually over the same range. This robustness stems from two properties: saliency estimation operates on hierarchical embeddings that integrate multi-scale context, and cosine similarity is invariant to magnitude scaling, making boundary detection insensitive to additive noise. Notably, Dywave's token count increases adaptively under noise, reflecting dynamic allocation of representational capacity as signal uncertainty grows. This behavior contrasts with fixed-patch methods, whose token count is fixed regardless of signal quality.

## 5. Related Work

**Human-Centric Sensing Signal Modeling.** Deep learning has driven substantial progress in human-centric sensing applications, including human activity recognition (Chen et al., 2023; Zhang et al., 2025), healthcare monitoring (Ren et al., 2025; Saha et al., 2025; Chatterjee et al., 2020; Ullah et al., 2022; Englhardt et al., 2024), and stress or affect recognition (Yu & Sano, 2023; Nath et al., 2023; Wang et al., 2023a). The sensing signals underlying these tasks are highly heterogeneous, non-stationary, and context-dependent, motivating extensive research on specialized model architectures (Yao et al., 2017; Ekambaram et al., 2023; Kara et al., 2024b; Shams et al., 2024) and learning frameworks (Deldari et al., 2022; Kimura et al., 2025; Ouyang et al., 2022; Zhang et al., 2022). Despite these advances, effectively transforming raw sensing streams into input representations that respect the intrinsic temporal structure of physical events remains a core challenge. Most existing approaches rely on predefined segmentation rules or external supervision, which limits their ability to adapt to irregular and event-driven temporal dynamics commonly observed in real-world sensing

scenarios (Zheng et al., 2025; Gao et al., 2023).

**Time-Series Tokenization and Representation.** Tokenization serves as a key interface between sensing signals and sequential models. A widely used strategy applies fixed-length windowing to partition time-series into uniform tokens, analogous to patching in vision transformers (Dosovitskiy et al., 2020; Nie et al., 2023; Cao et al.; Chang et al., 2025; Das et al., 2024; Jin et al.; Zhou et al., 2023). While simple and effective, fixed patching requires careful tuning of window size and stride and is inherently insensitive to variations in signal dynamics. Recent work explores multi-scale patching to capture temporal patterns at multiple resolutions (Cao et al., 2025; Zou et al., 2024; Zhong et al., 2024; Wang et al., 2024), though such methods are often tailored to forecasting and assume relatively homogeneous temporal structures. Other approaches discretize continuous signals into symbolic tokens inspired by language modeling (Sennrich et al., 2016; Ansari et al.; Masserano et al., 2025; Götz et al., 2025), but quantization can disrupt fine-grained temporal coherence critical for sensing data. Frequency-based representations further enrich modeling via time–frequency transforms (Yao et al., 2019; Kara et al., 2024a; Hu et al., 2025; Piao et al., 2024; Yi et al., 2023), yet their reliance on fixed windowing constrains temporal adaptivity. A key challenge is adapting tokenization granularity to match signal variability. LightGTS (Wang et al., 2025) learns per-sequence patching, enabling inter-sequence adaptation, but still overlooks intra-sequence heterogeneity. In vision, TokenLearner (Ryoo et al., 2021) and DynamicViT (Rao et al., 2021) perform adaptive token selection on tokenized patches and require modifications to the backbone. In contrast, sensing signals lack semantically meaningful token units, and tokenization must be constructed from continuous waveforms. Moreover, Dywave is decoupled from the backbone as a modular front-end that can be paired with arbitrary sequence models without architectural changes.

## 6. Conclusion

We introduced Dywave, a dynamic wavelet-based framework for adaptive time-series tokenization in sensing applications. By aligning token boundaries with intrinsic event transitions rather than fixed temporal windows, Dywave constructs representations that better reflect the underlying physical structure of sensing signals. Our results demonstrate that adaptive, event-aligned tokenization improves robustness and generalization across diverse sensing tasks and temporal conditions. This work highlights the importance of physically grounded, adaptive input representations as a foundation for scalable and semantically meaningful learning in human-centric sensing systems. Due to space limitations, we provide additional evaluation results and discussions in Appendix.

## Acknowledgements

Research reported in this paper was sponsored in part by the Army Research Laboratory under Cooperative Agreement W911NF-17-20196, NSF CNS 20-38817, and the Boeing Company. The views and conclusions contained in this document are those of the author(s) and should not be interpreted as representing the official policies of the CCDC Army Research Laboratory or the US government. The US government is authorized to reproduce and distribute reprints for government purposes, notwithstanding any copyright notation hereon.

## Impact Statement

This work addresses the tokenization gap between **continuous sensing signals** and **discrete input representations** for deep learning and Edge AI. We discuss below both the potential benefits and societal considerations.

**Positive Impacts.** We hope this framing brings attention to an important yet underexplored challenge at the intersection of sensing and machine learning. As AI systems increasingly operate in and interact with the physical world, bridging this gap becomes essential for developing models that can perceive and reason about continuous sensor streams. By highlighting this problem and proposing a principled solution, we hope this framing advances interdisciplinary research at the intersection of sensing and AI.

Additionally, Dywave's adaptive tokenization reduces the need for extensive hyperparameter tuning required by uniform patching methods, improving accessibility for practitioners. The event-aligned boundaries further contribute to automatic micro-activity segmentation, substantially reducing the annotation burden associated with fine-grained labeling in sensing applications, where raw signals are difficult to interpret and require manual effort. This automation also carries a direct privacy benefit. Since segmentation no longer requires human annotators to inspect raw sensor streams, sensitive physiological or behavioral recordings need not be exposed during the labeling process.

**Societal Considerations.** The sensing applications enabled by this work could potentially raise considerations common to ubiquitous computing. For human activity recognition, improved models could enhance assistive technologies for elderly care and rehabilitation monitoring, but similar capabilities could potentially be misused for surveillance without consent. For healthcare applications using ECG or physiological signals, more accurate monitoring benefits patient care while requiring careful attention to data privacy. Dywave's event-aligned tokenization produces compact representations that selectively retain semantically salient segments and discard stationary or uninformative intervals, which reduces the fidelity of stored data and can limit re-identification from retained representations.

We would like to note that Dywave is a general-purpose tokenization framework and does not itself collect or store personal data. All datasets used in this work are publicly available and no additional IRB approval was required for our experiments. As with many sensing-based systems, real-world deployment requires appropriate consent and adherence to privacy regulations, which are beyond the scope of this work.

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

# Appendix

The appendix of this paper is structured as follows.

- Section A details the evaluated datasets and statistics.

- Section B describes the baselines and the backbone encoders.

- Section C elaborates on the methodology.

- Section D specifies the implementation and training details.

- Section E provides additional evaluation results.

- Section F presents an additional case study.

- Section G discusses the limitations and future works.

*Table 6.* Dataset Statistics. Acc stands for Accelerometer, ECG for Electrocardiogram, and EMG for Electromyogram.

| Dataset Name | Classes | Modalities (Frequency) | Channel Size | Sample Length | # Samples |
|---|---|---|---|---|---|
| Ego4D (Grauman et al., 2022) | 10 | Acc (100 Hz) | 1 | 3000 (30 seconds) | 98315 |
| MOD (Liu et al., 2023) | 7 | Audio (8000 Hz), Seismic (100 Hz) | 1 | Audio: 16000, Seismic: 200 (2 seconds) | 8828 |
| PAMAP2 (Reiss & Stricker, 2012) | 18 | Acc (100 Hz), Gyroscope (100 Hz) | 3 | 360 (2 seconds) | 7075 |
| RWHAR (Sztyler & Stuckenschmidt, 2016) | 8 | Acc (100 Hz), Gyroscope (100 Hz) | 3 | 450 ( 5 seconds) | 12888 |
| WESAD (Schmidt et al., 2018a) | 8 | ECG (700 Hz), EMG (700 Hz) | 1 | 2100 ( 5 seconds) | 11229 |
| Ego4D-S (Grauman et al., 2022) | 10 | Acc (100 Hz) | 3 | 500 (5 seconds) | 59390 |
| MOD-OOD (Liu et al., 2023) | 4 | Audio (8000 Hz), Seismic (100 Hz) | 1 | Audio: 16000, Seismic: 200 (2 seconds) | 35162 |

# A. Dataset and Preprocessing

This section provides details on the datasets and preprocessing used in the experiments. Table 6 shows the dataset statistics.

## A.1. Datasets

- **Ego4D** (Grauman et al., 2022) is a large-scale egocentric dataset containing 836 hours of IMU recordings collected across 74 locations in 9 countries. We use the 100 Hz accelerometer signals covering 10 complex activities (*e.g.*, , cleaning, crafting, cooking) and segment each sequence into 5-second and 30-second clips for long-context evaluation and cross-sequence generalization. Data are randomly split by sequence into training, validation, and test sets following a 70:15:15 ratio.

- **Moving Object Detection (MOD)** (Liu et al., 2023) is a public dataset of seismic and acoustic signals collected from diverse moving objects across multiple environments, designed for nearby object classification that can enhance human-safety and situational awareness. For example, roadside sensing nodes can detect approaching vehicles or objects near intersections and pedestrian zones, offering early alerts to drivers and pedestrians. We segment each signal into 2-second samples, using 100 Hz seismic data for short-context evaluation and 8 kHz acoustic data for long-context evaluation. The dataset also provides an out-of-distribution dataset with different object types and environments, which we use to assess cross-domain generalization.

- **Physical Activity Monitoring (PAMAP2)** (Reiss & Stricker, 2012) is an open dataset for human activity recognition, collected from 9 subjects wearing three IMUs on the chest, wrist, and ankle. It includes 18 physical activities, such as walking, running, and cycling. For our experiments, we use the 100 Hz accelerometer and gyroscope signals, segmented into 2-second samples for short-context evaluation across both modalities.

- **Real World Human Activity Recognition (RWHAR)** (Sztyler & Stuckenschmidt, 2016) is a public IMU dataset collected from 15 subjects performing 8 everyday activities, such as stair climbing, jumping, and walking. Sensors were placed on seven body locations, and we use the accelerometer and magnetometer signals from the waist, segmented into 9-second samples. Ten subjects are used for training, two for validation, and the remainder for testing. Both modalities are evaluated under short-context settings.

- **Wearable Stress and Affect Detection (WESAD)** (Schmidt et al., 2018a) is a multimodal physiological dataset for stress and affect recognition, collected from 15 subjects using a chest-worn RespiBAN and a wrist-worn Empatica E4. It includes ECG, EDA, EMG, respiration, BVP, body temperature, and IMU signals. For our experiments, we use the 700 Hz ECG and EMG data from the RespiBAN, segmented into 3-second samples labeled as *stress* or *amusement*. Subjects are randomly divided into training (11), validation (2), and test (2) groups.

### A.2. Preprocessing

Each sample is a time-series segment of shape $C \times L$ where $C$ is the number of channels and $L$ the sequence length. During training, one augmentation is randomly selected from a pool of time-domain transformations, including permutation, scaling, negation, horizontal flip, time warping, and magnitude warping. The augmentation is then applied with a probability of 0.5 to enhance variability and generalization.

## B. Baselines

### B.1. Tokenization Baselines

To isolate the effect of tokenization, we restrict our comparisons to baselines that do not modify the underlying backbone architectures. All methods operate on the same sequence encoders with identical model configurations, differing only in how input signals are tokenized or segmented. This ensures a fair and controlled evaluation, where performance differences can be attributed to tokenization strategies rather than architectural changes or additional model capacity.

- **PatchTST** (Nie et al., 2023) tokenizes time-series inputs by segmenting them into fixed-length subseries, or *patches*, which serve as input tokens to the backbone encoder. Each univariate channel is patched and processed independently, with shared backbone weights across channels. The model relies on a fixed patch size and stride to control granularity and overlap between patches. We extensively tune these parameters and report the best-performing configuration.

- **DropPatch** (Qiu et al., 2025) follows the same fixed-size patching strategy as PatchTST, segmenting signals into subseries defined by preset patch size and stride. It randomly drops a portion of tokens during training to improve robustness and reduce overfitting. For fair comparison, we adopt the same optimal patch size and stride as PatchTST and fix the token drop rate at 0.5 across all experiments.

- **MedFormer** (Wang et al., 2024) adopts a multi-granularity patching strategy to model temporal dependencies at multiple resolutions. It builds parallel token streams with varying patch sizes and strides to model fine- and coarse-grained temporal dynamics. For fair comparison, we use the optimal patch size and stride from PatchTST as the base configuration and set additional granularities to 0.5×, 1×, 2×, and 4× of the base values.

- **WaveToken** (Masserano et al., 2025) converts quantized wavelet coefficients into discrete token IDs as opposed to projecting patches into a continuous embedding space. By decomposing inputs via the discrete wavelet transform, it represents time-localized frequencies as quantized tokens, similar to language model tokens, for learning in a discrete, frequency-aware space. Due to the large vocabulary size introduced by wavelet quantization, we evaluate WaveToken only on short-context signals to mitigate memory overhead.

- **MultiPatchFormer** (Naghashi et al., 2025) introduces a multi-scale patch embedding strategy to capture temporal dependencies at different resolutions and cross-channel correlations across signal dimensions. Each time series is divided into multiple patch streams with distinct sizes and strides, enabling joint modeling of fine- and coarse-grained dynamics. Streams are embedded independently and projected into a shared feature space for cross-scale interaction. For consistency, we use the optimal patch size and stride from PatchTST as the base configuration and extend additional granularities to 0.5×, 1×, 2×, and 4× of the base values.

- **LightGTS** (Wang et al., 2025) is a general time series forecasting architecture that handles diverse sampling scales and intrinsic frequencies during multi-source pre-training. It employs periodical tokenization, which adaptively divides each sample into patches based on cycle length. .

### B.2. Backbone Encoders

- **Transformer** (Vaswani et al., 2017) has been widely used for time-series modeling. It processes sequential inputs through multi-head self-attention layers, capturing long-range dependencies and contextual relationships between

---

**Algorithm 1** Dywave: Dynamic Tokenization Pipeline

---

**Require:** Raw signal $X \in \mathbb{R}^{C \times L}$, compression ratio $\tau$, partition index $K$, downsampling factor $s$

1: **Step 1: Wavelet Decomposition**
2: $\{dX_1, \ldots, dX_J, A\} \leftarrow \text{MODWT}(X)$
3: **Step 2: Hierarchical Embedding**
4: $X^U \leftarrow \text{Concat}(X, dX_1, \ldots, dX_K)$ {Detail stream}
5: $X^V \leftarrow \text{Concat}(dX_{K+1}, \ldots, dX_J, A)$ {Context stream}
6: $E^U \leftarrow \text{Conv1D}(X^U)$
7: $E^V \leftarrow \text{Interpolate}(\text{TransformerBlock}(\text{AvgPool1D}(\text{Linear}(X^V), s)), L)$
8: $E^F \leftarrow \text{Concat}(E^U, E^V)$
9: **Step 3: Temporal Anchor Formation**
10: $P_t \leftarrow 1 - \text{CosineSim}(F_k(E_{t-1}^F), F_q(E_t^F))$ for $t \in [2, L]$
11: $\mathcal{A} \leftarrow \text{TopK}(\text{NMS}(P, w_{\text{nms}}), \lceil \tau \cdot L \rceil)$ $\{w_{\text{nms}} = \lfloor L/(2\lceil \tau L \rceil) \rfloor\}$
12: **Step 4: Dynamic Temporal Fusion**
13: $\kappa(t) \leftarrow \arg\min_{a \in \mathcal{A}} |t - a|$ for $t \in [1, L]$
14: $E_k^A \leftarrow \sum_{t:\kappa(t)=a_k} P_t \cdot E_t^F / (\sum_{t:\kappa(t)=a_k} P_t + \varepsilon)$ for $k \in [1, |\mathcal{A}|]$
15: $E \leftarrow \text{MLP}(E^A)$
16: **Step 5: Reconstruction Loss (Training Only)**
17: $\hat{W} \leftarrow \text{AdaptivePool}(\text{ConvTranspose1D}(\text{Linear}(E)), L)$
18: $\mathcal{L}_{\text{rec}} \leftarrow \text{MSE}(\{dX_1, \ldots, dX_J, A\}, \hat{W})$
19: **return** $E, \mathcal{L}_{\text{rec}}$

---

tokens. For time-series data, each patch or segment is treated as an input token, and positional encoding is added to preserve temporal ordering.

- **Mamba2** (Dao & Gu, 2024) is a recently developed state-space model designed for efficient long-sequence modeling. Unlike attention-based architectures with complexity growing quadratically with the sequence length, Mamba2 employs a recurrent state-space formulation with linear computational complexity for scalable inference over extended time horizons. It models temporal dependencies via continuous-time dynamics and selective state updates, efficiently capturing both local and long-range signal patterns.

## C. Methodology

Dywave transforms raw sensing signals into compact, event-aligned token embeddings through a five-stage pipeline. Given an input $X \in \mathbb{R}^{C \times L}$, the framework first applies wavelet decomposition to extract multi-resolution coefficients capturing both high-frequency transients and low-frequency trends. These coefficients are then partitioned into detail and context streams, which are encoded via separate pathways and fused into per-timestep embeddings. A learned saliency function identifies temporal anchors at semantic transitions, and neighboring timesteps are aggregated into anchor-aligned tokens through saliency-weighted fusion. An auxiliary reconstruction objective ensures the compressed tokens preserve multi-scale signal structure. Algorithm 1 summarizes the complete pipeline.

### C.1. Signal Decomposition through Maximal Overlap DWT

To capture the temporal structure of time-series signals, we apply the *Maximal Overlap Discrete Wavelet Transform* (MODWT) (Percival & Walden, 2000) to decompose raw inputs into multi-resolution *context* and *details* components. Unlike the standard DWT, which downsamples the signal and imposes constraints on sequence length (Larrubia et al., 2025), MODWT is *undecimated*, preserving the original sequence length across all scales. This ensures temporal alignment between coefficients and the raw signal, producing a coherent multi-frequency *snapshot* at each timestamp. Formally, for an input $X \in \mathbb{R}^{C \times L}$ with $C$ channels and length $L$, MODWT computes wavelet and scaling coefficients at each scale $j \in [1, J]$ and timestep $t \in [1, L]$ as:

$$dX_{j,t} = \widetilde{W}_{j,t} = \sum_{l=0}^{L_j-1} \widetilde{h}_{j,l} \, A_{j-1,t-l \bmod L}, \qquad A_{j,t} = \widetilde{V}_{j,t} = \sum_{l=0}^{L_j-1} \widetilde{g}_{j,l} \, A_{j-1,t-l \bmod L}, \qquad A_{0,t} = X_t. \qquad (10)$$

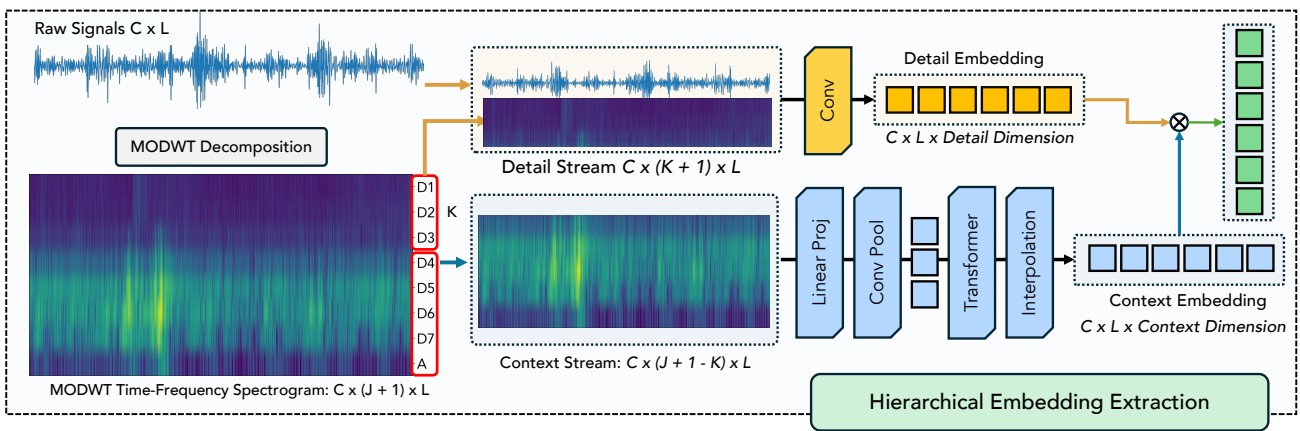

*Figure 10.* Physics-Informed Hierarchical Embedding Module.

Here, $\widetilde{W}_j$ denote the *discrete* wavelet coefficients at level $j \in [1, J]$, $A_j$ the corresponding *approximations*, $\widetilde{h}_j$ and $\widetilde{g}_j$ the rescaled wavelet and scaling filters, and $L_j$ the effective filter length. Since MODWT is undecimated, both $dX_j$ and $A_j$ preserve the full temporal resolution of the original sequence $L$.

The recursive formulation produces a hierarchy of approximations $\{A_j\}_{j \in 1, \dots, J}$ and details $\{dX_j\}_{1, \dots, J}$. We retain the discrete coefficients at every level and the coarsest approximation, yielding the MODWT output:

$$\{dX_1, dX_2, \dots, dX_J, A\} \in \mathbb{R}^{(J+1) \times C \times L} = \text{MODWT}(X), \tag{11}$$

MODWT disentangles fine transients from coarse contextual structure while maintaining precise temporal alignment across all scales, producing an interpretable, time-consistent hierarchy of signal representations for arbitrary sequence lengths $L$.

We leverage PyWavelets (Lee et al., 2019) as the MODWT implementation.

### C.2. Hierarchical Embedding

Figure 10 illustrates the hierarchical embedding module. Given the MODWT decomposition, we partition coefficients into detail and context streams based on frequency characteristics. The detail stream $X^U = \{X, dX_1, \dots, dX_K\}$ captures high-frequency transients, while the context stream $X^V = \{dX_{K+1}, \dots, dX_J, A\}$ encodes low-frequency trends. We set $K = 1$ by default, assigning the most fine-grained signal component to the detail stream.

**Detail Embedding**: To capture the detail embedding, we leverage convolution layers to extract fine-grained representations from the detail streams:

$$E^U = \text{Conv}(X^U), \quad E^U \in \mathbb{R}^{C, d_U, L}, \ X^U \in \mathbb{R}^{C \times (K+1) \times L}, \tag{12}$$

where $d_U$ is the detail embedding dimension projected from the detail stream.

**Context Embedding**: Unlike the detail stream, which benefits from localized convolution, modeling the context stream requires capturing long-range relationships across time and scale. To achieve this efficiently, we adopt self-attention within an *hourglass transformer* architecture (Nawrot et al.) that first compresses and then expands the temporal resolution. The context stream is projected into a latent space and adaptively downsampled according to the available computation budget:

$$X^V_{\text{down}} = \text{Conv1D}(\text{Linear}(X^V), \text{ stride} = s), \quad X^V \in \mathbb{R}^{C \times (J+1-K) \times L}, \ X^V_{\text{down}} \in \mathbb{R}^{C \times L^{\text{context}} \times d_V}, \tag{13}$$

where $d_V$ is the embedding dimension, and the stride $s$ is dynamically selected such that the downsampled length $L^{\text{context}} = L/s$ fits within the computation budget. A lightweight transformer encoder then models global dependencies over the shortened sequence, and the resulting context embedding is interpolated back to the original sequence length $L$ for alignment with the detail stream:

$$E^V = \text{Interpolation}(\text{Transformer}(X^V_{\text{down}}), \text{ size} = L), \quad E^V \in \mathbb{R}^{C \times L \times d_V}. \tag{14}$$

**Fusion of Embeddings**: Decomposing signals into detail and context streams provides a physics-informed representation that captures both instantaneous cues and global structure. For example, in IMU signals, the detail embedding $E^U$ can

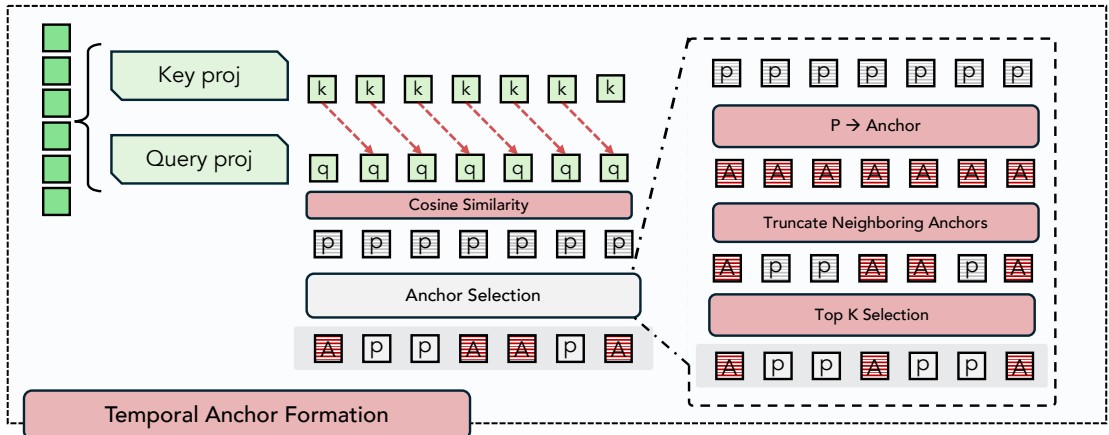

*Figure 11.* Temporal Anchor Formation Module.

capture abrupt, high-frequency transients such as wrist flicks or foot impacts, while the context embedding $E^V$ interprets these as transitions between broader activity phases, such as moving from walking to standing or from wiping to resting. To integrate these complementary views, we fuse the two embeddings into a unified hierarchical embedding:

$$E^F = E^U || E^V, \quad E^F \in \mathbb{R}^{C,L,d}; \ d = d_v + d_u. \tag{15}$$

### C.3. Temporal Anchor Formation

The hierarchical embeddings encode both fine-grained transients and coarse-grained contexts, providing multi-scale information for detecting event anchors. Rather than relying on fixed intervals, Dywave identifies anchors at semantic transitions in the embedding space. Figure 11 illustrates this module.

**Saliency Estimation via Key-Query Projections.** To detect transitions between events, each fused embedding is projected into key and query spaces through learned linear mappings:

$$k_t = F_k(E_t^F), \quad q_t = F_q(E_t^F), \quad F_k, F_q : \mathbb{R}^d \to \mathbb{R}^{d/4}. \tag{16}$$

The saliency at each timestep measures representational dissimilarity between adjacent frames:

$$P_t = 1 - \text{CosineSim}(k_{t-1}, q_t), \qquad t \in [2, L]. \tag{17}$$

Cosine similarity provides scale-invariant transition detection: continuous physical processes produce slowly varying embeddings with high similarity (low $P_t$), whereas genuine event transitions induce abrupt representational shifts that yield high saliency.

**Anchor Selection via NMS and TopK.** The raw saliency sequence may contain noisy or redundant peaks. To extract a compact set of anchors, we apply temporal non-maximum suppression followed by top-$k$ selection:

$$\mathcal{A} = \text{TopK}(\text{NMS}(P, w_{\text{nms}}), \lceil \tau \cdot L \rceil), \tag{18}$$

where $\tau \in (0, 1)$ is the target compression ratio controlling the maximum number of output tokens. NMS with window size $w_{\text{nms}} = \lfloor L/(2\lceil \tau L \rceil) \rfloor$ retains only the local maximum within each window, enforcing minimum separation between anchors.

In practice, NMS often produces fewer anchors than the budget $\lceil \tau \cdot L \rceil$, particularly for signals with sparse event transitions or long stationary intervals. The subsequent TopK operation serves as a safeguard against over-tokenization by capping the maximum token count to prevent computational explosion for signals with unusually dense saliency peaks. When NMS already yields fewer than the budget, TopK passes through all anchors.

### C.4. Dynamic Temporal Fusion

Once anchors are identified, Dywave consolidates embeddings within each segment into event-aligned representations. Real-world signals often contain long intervals of stable dynamics where consecutive timesteps convey similar semantics;

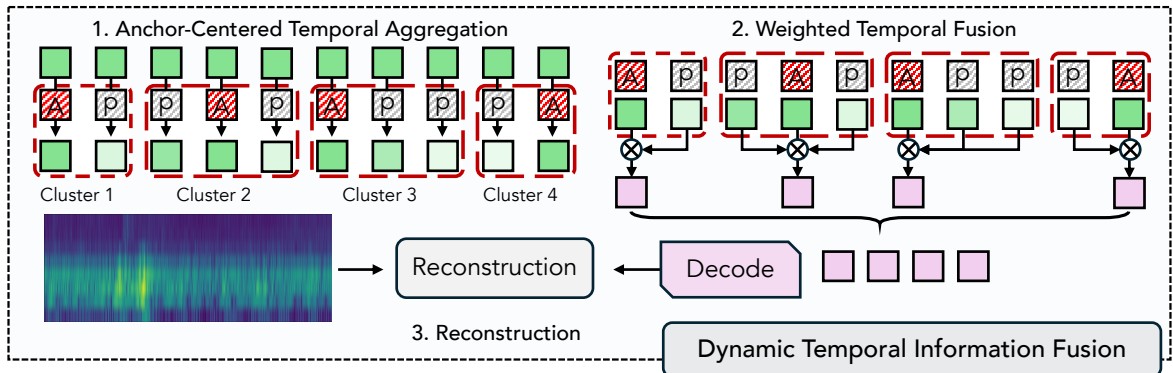

*Figure 12.* Temporal Fusion Module.

uniform patching wastes computation on such redundant information. Dynamic temporal fusion addresses this by adaptively compressing coherent regions while preserving semantic integrity at event boundaries. Figure 12 illustrates this module.

**Anchor-Centered Cluster Formation.** Each anchor corresponds to a semantically significant moment marking a shift in local dynamics. During fusion, anchors serve as cluster centers, with surrounding timesteps assigned to their nearest anchor:

$$\kappa(t) = \arg\min_{a \in \mathcal{A}} |t - a|, \qquad t \in [1, L]. \tag{19}$$

This induces a partition of the sequence into $|\mathcal{A}|$ contiguous segments with $O(L)$ complexity. Each segment groups temporally coherent timesteps that share similar dynamics, forming natural clusters aligned with signal semantics.

**Saliency-Weighted Aggregation.** Within each cluster, embeddings are aggregated using saliency scores as weights:

$$E_k^A = \frac{\sum_{t:\kappa(t)=a_k} P_t \cdot E_t^F}{\sum_{t:\kappa(t)=a_k} P_t + \varepsilon}, \qquad k \in [1, |\mathcal{A}|], \tag{20}$$

where $\varepsilon = 10^{-6}$ ensures numerical stability. This weighting scheme emphasizes timesteps with higher saliency, which correspond to event boundaries and informative transitions, while down-weighting redundant intervals with low saliency. As a result, the fused embedding captures distinctive boundary characteristics while compactly representing stationary regions.

The aggregated embeddings are projected to the final token embedding space via a two-layer MLP:

$$E = \text{MLP}(E^A), \quad E \in \mathbb{R}^{|\mathcal{A}| \times d}, \tag{21}$$

where $d$ is the final token embedding dimension equal to the hidden dimension of the backbone encoder.

### C.5. Reconstruction Decoder

Dynamic fusion compresses variable-length segments into fixed-dimensional token embeddings, which risks discarding fine-grained signal characteristics. To regularize this compression and ensure fused tokens preserve multi-scale structure, Dywave employs an auxiliary reconstruction objective during training.

**Decoder Architecture.** The decoder reconstructs MODWT coefficients from compressed tokens through three stages:

$$\hat{W} = \text{AdaptivePool}(\text{ConvTranspose1D}(\text{Linear}(E)), L) \in \mathbb{R}^{(J+1) \times C \times L}. \tag{22}$$

First, a linear layer projects each token embedding from dimension $d$ to an intermediate dimension matching the number of wavelet levels times channels: $\text{Linear} : \mathbb{R}^d \to \mathbb{R}^{(J+1) \cdot C}$. Second, a transposed 1D convolution expands the temporal dimension, producing an initial reconstruction at length proportional to $|\mathcal{A}|$. Finally, adaptive average pooling resamples to the original sequence length $L$, ensuring temporal alignment with the MODWT coefficients regardless of compression ratio.

**Wavelet Supervision.** Rather than reconstructing the raw signal directly, we supervise in the wavelet coefficient domain:

$$\mathcal{L}_{\text{rec}} = \text{MSE}(\{dX_1, \ldots, dX_J, A\}, \hat{W}). \tag{23}$$

This formulation offers two advantages. First, it explicitly enforces preservation of both high-frequency transients (via detail coefficients $\{dX_j\}$) and low-frequency trends (via approximation $A$), preventing the model from collapsing to smooth reconstructions that ignore rapid dynamics. Second, supervising at multiple scales provides richer gradient signals than raw signal MSE, which can be dominated by low-frequency components.

**Training and Inference.** The reconstruction loss is weighted by $\lambda_{\text{rec}} = 0.1$ and combined with the task loss: $\mathcal{L} = \mathcal{L}_{\text{task}} + \lambda_{\text{rec}} \cdot \mathcal{L}_{\text{rec}}$. At inference, the decoder is discarded entirely, adding no computational overhead to the forward pass. The reconstruction objective thus serves as a regularizer that encourages information-preserving compression.

# D. Training and Implementation

## D.1. Training Details

All models are implemented in PyTorch 2.6.0 and trained on a single NVIDIA A5000 GPU with 24GB memory. We use the Adam optimizer (Kingma & Ba, 2014) with an initial learning rate of $1 \times 10^{-4}$ and a cosine annealing scheduler that decays the learning rate to $1 \times 10^{-6}$ over the training period. Models are trained for 500 epochs. The batch size is set to 256 for short-context datasets and 64 for long-context datasets to accommodate memory constraints.

## D.2. Generalization Finetuning Details

For cross-domain and sequence-length generalization experiments, we use two finetuning strategies. In *full backbone finetuning*, the tokenization module is frozen while the encoder and classification head are updated with a reduced learning rate of $1 \times 10^{-5}$. In *head-only finetuning*, both the tokenization module and encoder are frozen, and only the classification head is trained with the same learning rate. Both strategies use the Adam optimizer with cosine scheduler.

# E. Evaluation

## E.1. Ablation Studies - Variants Descriptions

We evaluate Dywave with five variants by removing individual components or replacing the original design of Dywave with alternative implementations:

- **Dywave-w/oWave** retains the anchor formation and dynamic fusion modules but replaces Dywave's hierarchical embedding module with PatchTST, isolating the hierarchical decomposition.

- **Dywave-FixedDWT** applies the standard MODWT on PatchTST using the same optimal patch size and stride as PatchTST, dropping Dywave's hierarchical embeddings and adaptive segmentation to evaluate the effect of fixed-scale wavelet decomposition.

- **Dywave-w/oRecon** removes the reconstruction objective $\mathcal{L}_{\text{rec}}$ and trains the model only with the backbone task loss to examine the influence of the reconstruction branch on representation quality.

- **Dywave-w/oFusion** uses only the anchors from the anchor formation module without performing the subsequent fusion, allowing us to evaluate the role of temporal fusion module.

- **Dywave-CNNBound** replaces the similarity-based anchor formation module with convolutional layers followed by a sigmoid probability head to analyze the context-aware similarity module.

- **Dywave-SpecBound** replaces cosine similarity with spectral energy as the saliency criterion. Specifically, the saliency score at each timestep is computed as the $\ell_2$ norm of the wavelet detail coefficients in the frequency domain, rather than the cosine dissimilarity between adjacent hierarchical embeddings. This isolates the contribution of semantic-space saliency by substituting a signal-level spectral measure that does not depend on the learned representation.

## E.2. Computation Efficiency

We also evaluate how the compact input tokens generated by Dywave can translate to real-world deployment efficiency. Table 7 shows the inference latency breakdown on a Raspberry Pi 4 device using input tokens generated by Dywave and the fixed-size PatchTST baseline. Across all datasets, Dywave consistently achieves lower latency due to the substantial

| Model | Component | Ego4D (30s) | PAMAP2 (5s) | RWHAR (4.5s) | MOD (2s) |
|---|---|---|---|---|---|
| Dywave | DWT | 0.0041 | 0.0018 | 0.0023 | 0.0014 |
| | Encode | 0.1125 | 0.0666 | 0.0414 | 0.0180 |
| | Saliency | 0.0512 | 0.0098 | 0.0126 | 0.0057 |
| | Merge | 0.0199 | 0.0096 | 0.0135 | 0.0087 |
| | Backbone | 0.0640 | 0.0401 | 0.0616 | 0.0705 |
| PatchTST | Patch | 0.0021 | 0.0013 | 0.0025 | 0.0012 |
| | Backbone | 0.3977 | 0.1558 | 0.2028 | 0.0832 |

*Table 7.* **Transformer** backbone encoder inference latency on the Raspberry Pi 4 device.

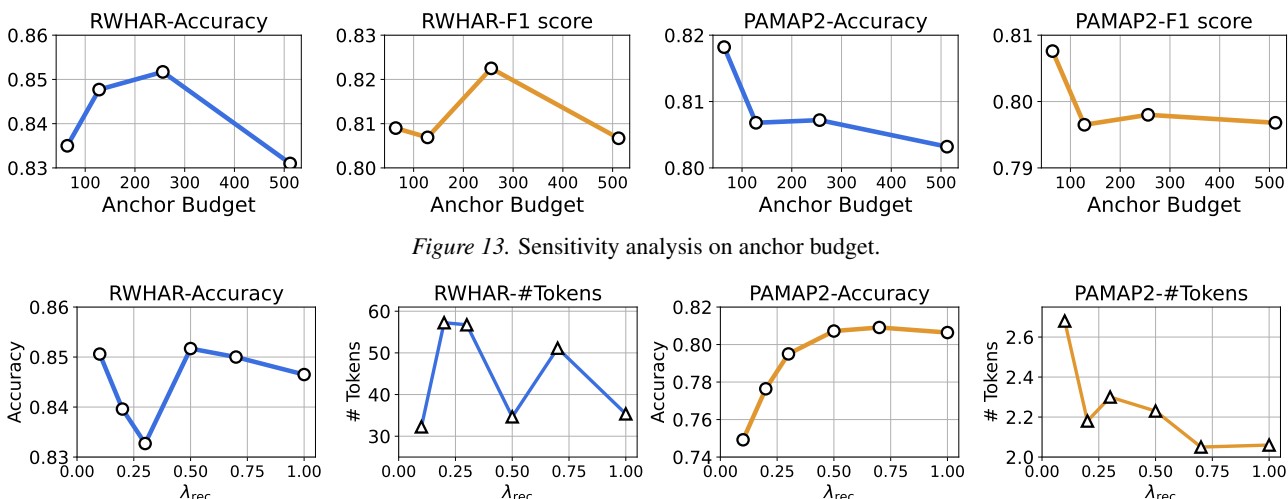

*Figure 13.* Sensitivity analysis on anchor budget.

*Figure 14.* Sensitivity analysis on reconstruction loss $\lambda_{\text{rec}}$.

reduction in input sequence length. The gap is more significant with long-context inputs in Ego4D. In these scenarios, PatchTST produces hundreds of tokens, leading to rapidly increasing latency with sequence length. In contrast, Dywave adaptively compresses long stationary regions into a small number of semantically coherent input tokens with up to an order-of-magnitude reduction in token count and a correspondingly lower inference delay of the backbone encoder.

### E.3. Sensitivity Analysis

Figures 13 and 14 report sensitivity to the maximum anchor budget $\tau$ and the reconstruction loss weight $\lambda_{\text{rec}}$. Performance remains mostly stable across anchor budgets from 64 to 512. Since NMS and the learned saliency landscape together frequently yield far fewer anchors than the maximum, the budget serves as a safe upper bound rather than directly determining token boundaries. Performance is similarly stable across $\lambda_{\text{rec}} \in [0.1, 1.0]$, and token count is non-monotonic in $\lambda_{\text{rec}}$, confirming that reconstruction regularizes representation quality without inflating anchor density. Fixed-patch methods such as PatchTST require extensive grid search over patch size and stride, as these directly determine tokenization structure. Dywave's is stable across all five evaluation datasets without extensive per-domain tuning, in contrast to PatchTST's search for patch sizes that could fluctuate the performance.

## F. Case Study on Human Activity Recognition

### F.1. Visualization Analysis

We perform a visualization study to examine how Dywave's learned boundaries align with the semantics of human activity events. Figure 15 presents boundary visualizations on long-sequence samples from the Ego4D accelerometer dataset under varying scenarios. The top row shows annotated motion events with red bounding boxes to illustrate the correspondence between the dynamic regions and boundaries generated by Dywave.

Even when the **same user performs the same activity** (*e.g.*, cleaning), the motion patterns differ substantially across samples due to variations in style, duration, and contextual behavior (row 1). This variability becomes more significant when

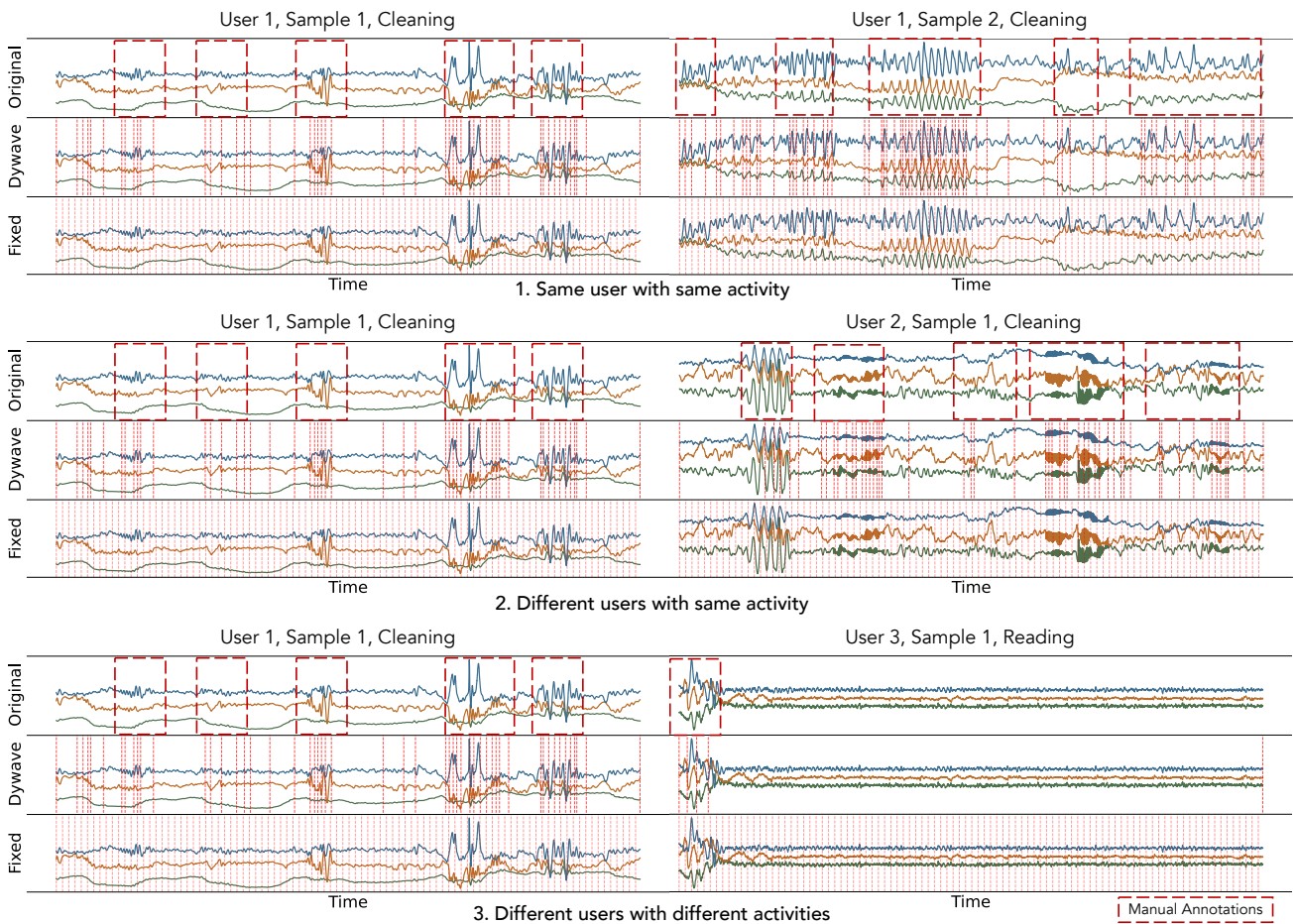

*Figure 15.* Ego4D Boundary Visualization. Signal events are manually **annotated with red bounding boxes**.

**different users perform the same activity** (row 2), with user-dependent rhythms and intensities. Comparing **different activities** (row 3) further highlights changes in temporal density and dynamic range, reflecting the inherent heterogeneity of real-world motion across the samples.

Under such diverse conditions, tokenization with fixed-size patching fails to preserve semantic alignment and allocates equal granularity to both active and quiescent intervals. In comparison, Dywave dynamically adapts segmentation granularity, as high-motion regions result in fine-grained tokens while stable regions are compactly represented. For static activities such as reading, PatchTST produces redundant tokens over long stationary periods, whereas Dywave condenses them into a single event-level token and focuses on transient motion bursts.

These visualizations highlight how Dywave adapts to the inherent temporal heterogeneity of human activity signals, generating event-aligned representations that remain semantically coherent across users, contexts, and motion patterns. The boundaries produced by Dywave provide practitioners with a clear structure for analyzing non-intuitive raw sensing signals and demonstrate efficiency for downstream tasks with minimal overhead.

### F.2. Human-Centric Micro-Activity Decomposition

Real-world human activities are composed of numerous short, fine-grained motions such as *reaching*, *grabbing*, or *walking* that sequentially form higher-level tasks. However, due to the annotation inefficiency and the non-intuitive nature of raw sensor signals, most datasets provide only coarse-grained activity labels (*e.g.*, *cooking*, *cleaning*), omitting these transient micro-activities. This lack of fine-grained labeling limits our ability to understand how humans perform these tasks and how transitions occur between motions. Manual annotation of such micro-activities is time-consuming and error-prone, since many transitions occur in sub-second intervals and lack clear visual or temporal boundaries in the sensor space. Consequently, a deeper, structure-aware understanding of human behavior through sensing signals has remained challenging. This section

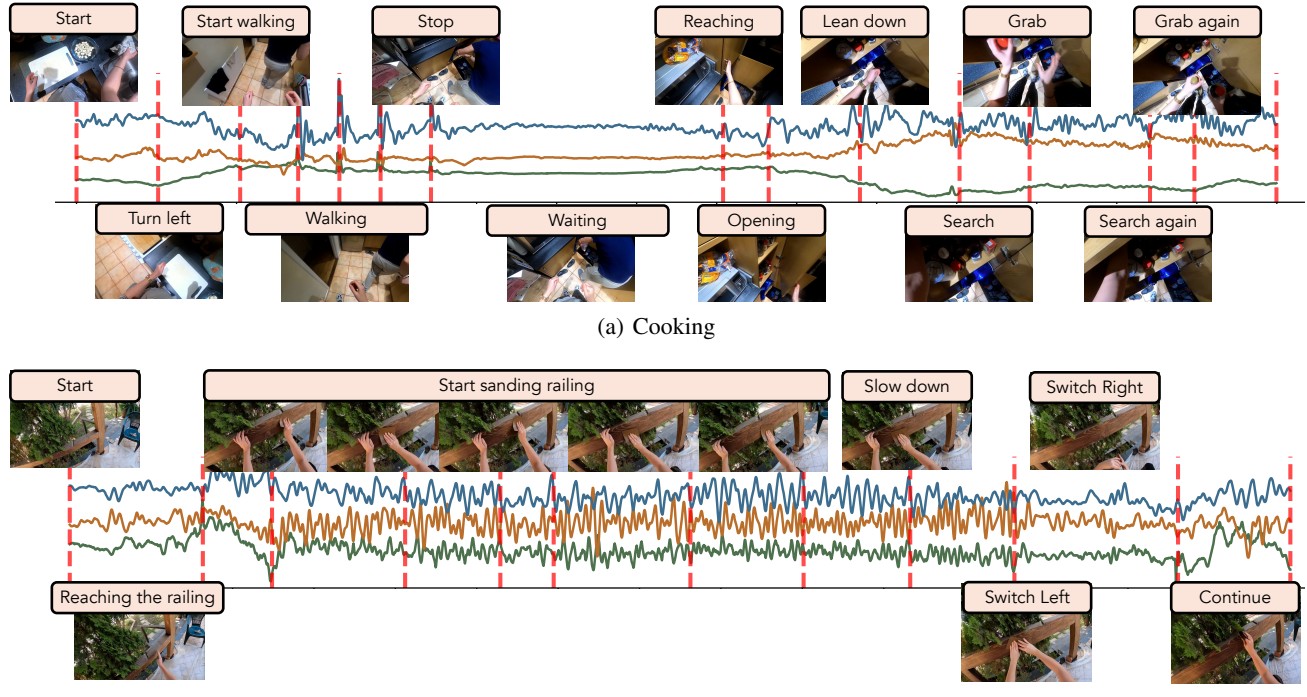

(a) Cooking

(b) Household Management

*Figure 16.* Example of micro-activity decomposition with Dywave on Ego4D.

conducts a qualitative case study of Dywave's capability in mitigating this challenge on the Ego4D dataset (Grauman et al., 2022), which provides synchronized egocentric video and IMU signals during daily activities such as *cooking*, *crafting*, and *household management*. We extract 15-second continuous IMU segments and apply Dywave to the accelerometer signals to automatically generate event-aligned boundaries without any fine-grained supervision.

**Micro-Activity Identification:** Figure 16 shows two examples from *cooking* and *household management* tasks. For each detected segment, we display the corresponding video frame and manually annotate the micro-activity being performed. Dywave's dynamic boundaries align closely with perceptible changes in human motion, such as *reaching for the cabinet door*, *grabbing a bottle*, or *start and end walking*. These short, contextually meaningful actions are not labeled in the original dataset but are automatically reflected by Dywave's boundary detection. Each resulting input token thus provides an intuitive and temporally localized representation of a micro-activity.

**Human-centric Micro-Activity Decomposition:** Dywave introduces a human-centric capability for understanding and interacting with human behavior at a more granular level. By transforming coarse, high-level signals into semantically coherent micro-activity units, it enables systems to learn the internal temporal structure of human actions without requiring expensive manual annotation. This decomposition can serve as an automatic proxy for fine-grained labeling, facilitate the construction of hierarchical activity taxonomies, and generate rich behavioral logs for long-term studies of human routines. In essence, Dywave shifts the focus of sensing intelligence from merely classifying what activity a person is doing to uncovering how it unfolds, through fine-grained sub-event detection and reasoning over complex temporal dynamics. Without such adaptive segmentation, these insights would remain hidden in continuous, unstructured IMU signals, underscoring the role of Dywave as a bridge between raw sensor data and human-understandable behavior representations.

## G. Discussion

Dywave demonstrates that adaptive, physics-informed tokenization can effectively bridge raw time-series sensing signals with semantically meaningful input representations. However, we recognize limitations of the current design and outline potential directions for future work.

**Computational Overhead:** While Dywave substantially improves encoder efficiency and performance at inference time by reducing redundant input length, this introduces additional computational overhead during preprocessing compared to fixed-size tokenization. Specifically, the wavelet decomposition and hierarchical embedding modules require extra

computations before the input tokens are passed to the backbone encoder. Although this cost is amortized by the reduction in input length for the heavy backbone encoder, the preprocessing module may still be significant for ultra-low-power or edge deployments. Future work could explore alternative techniques that maintain adaptivity with reduced overhead.

**Injecting Temporal Information:** While Dywave does not explicitly encode real-world elapsed time, its physics-informed hierarchical embeddings already capture relative temporal dynamics across multiple scales. Each token represents an event whose duration is implicitly reflected in the local continuity and wavelet coefficients. Thus, temporal information is indirectly preserved in the learned embeddings. Nonetheless, explicit modeling of absolute or physical time remains an open research direction. Future extensions could explore joint representations that align the model's internal time with human-understandable physical timescales, potentially improving synchronization and interpretability in real-world deployments.

**Dependency on Wavelet Basis:** The discrete wavelet transforms provide Dywave with a physics-informed structured view of the signals. However, the choice of the wavelet basis (*e.g.*, Daubechies, Symlets, Coiflets (Lina & Mayrand, 1995; Graps, 1995)) could yield different time-frequency trade-offs and potentially influence the quality of anchor selections across different sensing datasets and modalities. Although our experiments using Daubechies 4 (db4) wavelet basis show general robustness, we acknowledge that certain applications may require task-specific basis selection to better capture characteristic patterns. An interesting direction of future research is to explore learnable modules for the wavelet transform as an alternative that can adapt to the signals effectively and efficiently.

**Multimodal Interaction:** Although we have demonstrated Dywave's robustness when applied to multimodal data, it does not explicitly model cross-modal interactions. Dywave primarily focuses on unimodal signals, where adaptive tokenization is guided by temporal dynamics within each individual modality. Extending to multimodal sensing introduces additional challenges such as temporal synchronization and modality-specific phase alignment, which can serve either as informative cues or redundant patterns that may be selectively pruned. Extending Dywave to explicitly capture these multimodal dependencies is a potential direction for future work.

**Extension to Additional Tasks:** Dywave is designed for event-aligned representation learning in heterogeneous IoT sensing, where the primary task is classification. This limits Dywave as a tokenization framework for IoT sensing classification rather than a universal time-series method. Extending dynamic tokenization to time-series *forecasting* and *anomaly detection* is an interesting future direction. Additionally, the current framework requires task-labeled data to supervise the saliency landscape through the downstream objective. An important future direction is *self-supervised* training, where tokenization boundaries are learned jointly with the self-supervised objective without relying on task-specific annotations. This would enable domain-agnostic, label-efficient tokenization that adapts to diverse sensing modalities without per-task fine-tuning.

