# OpenReview forum: "Dywave: Event-Aligned Dynamic Tokenization for Heterogeneous IoT Sensing Signals"
_ICML.cc/2026/Conference — ICML 2026 regular_

### Official Review · Reviewer_BNsG · 2026-02-23

**Soundness:** 3
**Presentation:** 3
**Significance:** 3
**Originality:** 3
**Overall Recommendation:** 4
**Confidence:** 3

**Summary:**

This paper addresses the "tokenization gap" in IoT sensing, where traditional methods arbitrarily slice continuous signals into fixed windows, often disrupting the semantic integrity of physical events. To resolve this, the authors propose Dywave, a dynamic framework that adapts token creation to the signal's intrinsic content. The method leverages wavelet decomposition to separate the signal into multi-scale details and contextual trends, allowing it to capture both transient changes and long-term patterns. By calculating the similarity between these hierarchical representations, the model identifies significant transitions—termed, "anchors"—that mark the boundaries of physical events. Instead of uniform slicing, the system dynamically fuses time steps around these anchors to form compact, meaningful tokens. Experiments across diverse datasets demonstrate that this approach improves classification accuracy while significantly reducing the number of tokens required for the backbone model.

**Compliance With Llm Reviewing Policy:**

Affirmed.

**Final Justification:**

Thank you for the rebuttal and discussion. The clarifications on latency, memory, and hyperparameter sensitivity are helpful, and the evidence on longer, more heterogeneous signals is appreciated. However, I still have reservations about the added preprocessing overhead and overall complexity, which somewhat limit the practical impact. Therefore, I maintain my score of 4.

**Key Questions For Authors:**

End-to-End Latency: Could the authors clarify the total inference time, explicitly including the time taken for wavelet decomposition and anchor formation? It is important to understand if the preprocessing overhead outweighs the savings in the backbone model for real-time applications.

 Memory Usage: Given that the wavelet transform expands the input data dimensions significantly before compression, how does this affect the peak memory requirements during training and inference compared to standard methods?

Handling Stationary Signals: How does the method perform on biomedical signals where "quiet" or stable periods might contain critical physiological information? Is there a risk that the compression mechanism discards subtle but clinically relevant anomalies?

**Limitations:**

Yes.The authors acknowledge that the dynamic nature of the method introduces additional computational overhead during the preprocessing phase. They also note that the current implementation processes each channel or modality independently, without explicitly modeling the interactions between different sensors.

**Strengths And Weaknesses:**

Strengths
The paper is strongly motivated by the physical nature of sensing data, successfully integrating established signal processing techniques with modern deep learning. The use of wavelet decomposition provides a principled way to handle non-stationary signals, ensuring that multi-scale dynamics are preserved. A significant practical advantage is the substantial reduction in the sequence length fed into the backbone model—sometimes reducing token count by nearly four times—which directly lowers the inference cost of the heavy transformer layers(). Furthermore, the qualitative analysis reveals that the dynamically generated tokens align surprisingly well with human micro-activities, offering a level of interpretability and explainability that is typically absent in fixed-patching methods.

Weaknesses
Despite the claimed efficiency, the paper overlooks the computational cost of the preprocessing stage. The wavelet decomposition and the dynamic anchor selection process are computationally intensive operations that occur before the data reaches the backbone, potentially negating the speed gains for shorter sequences. Additionally, the method relies on sensitive hyperparameters to control the compression rate, which may require careful tuning for different domains to avoid losing critical information. While the reported "up to 12%" improvement is impressive, the gains on standard human activity recognition benchmarks are often marginal compared to simpler baselines, raising questions about whether the added architectural complexity is fully justified by the performance increase.

---

> ### Author Rebuttal · Authors · 2026-03-29
>
> We thank Reviewer BNsG for the careful and thoughtful review. We address each concern below.
>
> ---
> **W1 + W2 + Q1+ Q2. Computational cost**
>
> We provide a detailed wall-clock and memory comparison below. The answer depends on the sequence length and signal characteristics.
>
> |Dataset(Duration)|PatchTST Time(s)|Dywave Time(s)|PatchTST Mem(MB)|Dywave Mem(MB)|
> |-|-|-|-|-|
> |MOD Seismic (2s)|0.0886|0.1095|378.5|391.6|
> |RWHAR Acc (4.5s)|0.2151|0.1444|397.4|380.6|
> |PAMAP2 (5s)|0.1610|0.1330|391.7|374.0|
> |Ego4D (30s)|0.4102|0.2579|512.4|400.9|
>
> For seismic, preprocessing overhead dominates due to limited token reduction. Beyond ~4.5s, compression reduces both latency and memory, and the gap increases with longer contexts. The processing overhead depends on the information density. For dense signals, the preprocessing overhead exceeds the backbone savings because of limited redundancy to compress. For longer sequences with heterogeneous dynamics, Dywave's token compression substantially reduces backbone computation. Critically, as we scale to longer context windows or use larger encoder models, Dywave's advantage grows because backbone cost scales with semantics while preprocessing cost remains relatively fixed. This makes Dywave particularly well-suited for real-world deployments where signals are long, heterogeneous, and resource constraints are tight. More importantly, backbone computation can scale quadratically, so the efficiency advantage of Dywave increases with longer contexts and larger models.
>
> We also report peak memory (maximum resident set size) for the full pipeline. While MODWT can temporarily expand the representation before compression, this occurs only in the tokenization front-end. For longer signals, Dywave's memory usage is notably lower because the backbone dominates the total cost, and Dywave produces far fewer tokens. As sequence length grows, Dywave's memory advantage increases because the savings from fewer backbone tokens outweigh the temporary wavelet overhead.
>
> ---
> **W3. Sensitivity of anchor hyperparameter**
>
> We provide a hyperparameter sensitivity analysis below, covering the anchor budget ∈ [64,512] on RWHAR and PAMAP2.
>
> |Anchor Budget|RWHAR Acc|RWHAR F1|PAMAP2 Acc|PAMAP2 F1|
> |-|-|-|-|-|
> |64|0.8350|0.8090|0.8182|0.8076|
> |128|0.8477|0.8069|0.8068|0.7965|
> |256|0.8517|0.8225|0.8072|0.7980|
> |512|0.8310|0.8067|0.8032|0.7968|
>
> Performance is stable across the range. While PAMAP2 shows a wider spread at low λ_rec (where reconstruction is too weak to regularize compression), the overall sensitivity is substantially milder than fixed-patch methods. Unlike PatchTST, where patch size and stride directly determine tokenization structure and require per-domain grid search, Dywave's parameters act as mild regularizers on an adaptive process. The actual token boundaries and count are determined by learned saliency, not by these hyperparameters. The default settings work well across all five datasets without per-domain tuning, reducing the hyperparameter burden compared to uniform patching.
>
> ---
> **W4. Marginal gains**
>
> We address this from two perspectives.
>
> First, Dywave's contribution is not solely accuracy improvement but token efficiency. On benchmarks where accuracy gains are modest (e.g., short-context HAR), Dywave achieves comparable or better performance with significantly fewer tokens. For example, on PAMAP2 with Mamba2, Dywave uses ~2.23 tokens on average while achieving the best accuracy, a substantial reduction from the dozens or hundreds used by fixed-patch methods. This yields more compact representations and lower downstream cost.
>
> Second, Dywave's advantages become more pronounced in challenging settings. In long-context classification, it shows gains across long-context datasets. In cross-domain generalization, it maintains performance under domain shifts while producing far fewer tokens. In scalability, from 5s to 30s, PatchTST increases tokens by 6x, while Dywave grows modestly (15.5→35), indicating token density scales with semantic complexity rather than signal length.
>
> The added complexity is justified by producing compact, semantically meaningful token representations that improve scalability, generalization, and token efficiency.
>
> ---
> **Q3. Handling stationary medical signals**
>
> We acknowledge this concern. Dywave achieves competitive but not superior performance compared to PatchTST on WESAD, which we attribute to the relatively regular, periodic nature of these signals where uniform patching already provides reasonable coverage. Dywave does not discard stationary regions; it aggregates them into fewer tokens, preserving subtle variations within stationary periods in the fused representation. Because boundary segmentation is learned end-to-end, task-relevant stationary structures can still be captured when they matter. We will add a discussion of this trade-off, noting that Dywave is most beneficial for non-stationary, heterogeneous signals.

---

> > ### Author Rebuttal · Reviewer_BNsG · 2026-04-02
> >
> > Thank you for the detailed rebuttal. The added clarification on latency, memory, and hyperparameter sensitivity is helpful, and the discussion better explains the practical trade-offs of the method. I appreciate the evidence that Dywave is more effective for longer and more heterogeneous signals. That said, I still think the additional preprocessing overhead and overall method complexity somewhat limit the practical impact, so I will maintain my score of 4.

---

> > > ### Author Response · Authors · 2026-04-03
> > >
> > > Thank you for your response. We are glad that our clarification helped make the trade-offs more transparent.
> > >
> > > We understand your concern that, compared to simple heuristic approaches, our method introduces additional preprocessing overhead and complexity. However, we view this as a deliberate design choice of Dywave to prioritize representation quality and token efficiency, which are increasingly critical as backbone encoders scale across both edge and cloud deployments.
> > >
> > > Importantly, this overhead is amortized as sequence length grows, and in practical settings evaluated, Dywave achieves sub-second runtime on edge devices while providing consistent gains in downstream performance. We therefore believe the trade-off is justified, particularly for long-horizon sensing applications where token efficiency directly impacts feasibility.
> > >
> > > We sincerely appreciate your feedback and would be happy to further discuss any additional concerns you may have in the remaining rebuttal period.
> > >
> > > Best,
> > >
> > > Authors of Submission #6216

---

### Official Review · Reviewer_spNB · 2026-03-09

**Soundness:** 3
**Presentation:** 3
**Significance:** 2
**Originality:** 2
**Overall Recommendation:** 4
**Confidence:** 3

**Summary:**

This paper addresses the problem of tokenizing continuous, heterogeneous IoT sensing signals like ECG and EMG. The authors identify a "tokenization gap". Uniform fixed-size windows commonly used for time-series tokenization are misaligned "[...] with the intrinsic dynamic structures of physical events, which rarely conform to uniform timescales". Mainly, uniform windows can produce redundant tokens and can potentially segment the time series at unadvantageous positions.

Dywave addresses the tokenization gap by dynamically aligning tokens with events. It uses wavelet decomposition (MODWT) to capture multi-resolution features, computes saliency to detect important changes, selects certain moments as anchors, and merges adjacent steps. During training, it adds a reconstruction loss to improve token quality. Dywave works as a preprocessing step and is compatible with main-stream backbone encoders.

Dywave is tested on five real-world sensing datasets, covering short and long tasks, multimodal inputs, cross-domain generalization, and Raspberry Pi 4 deployment. Results show higher accuracy on some HAR tasks, fewer tokens, and reduced encoder latency, all while maintaining or improving F1 scores.

**Compliance With Llm Reviewing Policy:**

Affirmed.

**Final Justification:**

The authors rebuttal mostly addressed my concerns.
Therefore, I increased the score to 4.
However, I still see potential for a more detailed evaluation to better underline the claims.

**Key Questions For Authors:**

1) The authors show that Dywaves is especially useful for long-context inputs. The exact tradeoff between pre-processing overhead and the benefit of reduced sequence length could be explored to a larger extend. Specifically, for the examined backbones, is there a sequence length where the pre-processing overhead is larger than the gains from the reduced sequence length? If yes, where is the threshold.

2) For multimodal inputs, do you compute anchors jointly across concatenated embeddings or per-modality and then align? How often do anchors disagree across modalities, and how do you resolve conflicts?

3) How does the model compare to other non fixed window tokenization strategies like those presented in LightGTS or STaTS. The general concept has also been explored for other modalities (e.g., TokenLearner, DynamicViT). How does your approach fit into the related literature?

**Limitations:**

yes

**Strengths And Weaknesses:**

## Strengths
- The paper convincingly explains why uniform patching is a poor fit for heterogeneous signals (event fragmentation, redundant tokens, hyperparameter sensitivity).

- The method is well-motivated and modular, its components form a coherent pipeline.

- The paper includes a thorough empirical evaluation of the proposed approach, across diverse datasets.

- Ablation studies strengthen the claim that individual components contribute to performance.

- The paper is clearly written and sufficiently easy to follow.

## Weaknesses

- Major: Relevant prior research into non fixed window tokenization is omitted.

- Major: The authors claim Dywave segments signals into semantically meaningful events, but never objectively demonstrate this. Figure 12 provides some visual examples, but does not constitute a proper prove of the claim.

- When multiple modalities are used, it is not fully clear whether anchors are selected per-modality or jointly. The effect of per-modality misalignment and how Dywave handles it requires clarification and more experiments.

- Very minor remark: On every page, the manuscript mentions: "Title Suppressed Due to Excessive Size". This might be an issue with the template.

---

> ### Author Rebuttal · Authors · 2026-03-29
>
> We thank Reviewer spNB for the thoughtful and constructive review. We address each concern below.
>
> ---
> **W1 + Q3. Relevant prior research**
>
> We appreciate this important suggestion and have included comparisons with two non-fixed-window tokenization methods. LightGTS learns a variable patch size per sequence, allowing different per-sequence granularities. However, it applies uniform windows within each sample, overlooking intra-sequence heterogeneity where a signal may contain both rapid transient events and long stationary intervals that require different temporal resolutions. This is precisely the limitation Dywave addresses through adaptive tokenization. Ruptures applies online change-point detection to determine segmentation boundaries based on statistical signal properties.
>
> |Backbone|Method|RWHAR Acc|RWHAR F1|# Tokens|PAMAP2 Acc|PAMAP2 F1|# Tokens|
> |-|-|-|-|-|-|-|-|
> |Transformer|LightGTS|0.8586|0.8513|**14.66**|0.6830|0.6621|10.71|
> ||Ruptures|0.7761|0.7302|57.60|0.6877|0.6604|37.95|
> ||Dywave|**0.9094**|**0.8932**|29.61|**0.7977**|**0.7905**|**3.76**|
> |Mamba2|LightGTS|0.7917|0.7409|**14.66**|0.7003|0.6977|10.71|
> ||Ruptures|0.7998|0.7665|57.60|0.6629|0.6341|37.85|
> ||Dywave|**0.8517**|**0.8225**|34.65|**0.8072**|**0.7980**|**2.23**|
>
> Dywave outperforms both methods substantially across datasets and backbone architectures. While LightGTS uses fewer tokens, Dywave achieves substantially higher accuracy, indicating that the additional tokens carry meaningful semantic information that per-dataset uniform patching misses. We also note that Ruptures take much longer time during training and inference due to online running algorithm, which further highlight Dywave’s superior performance.
>
> Regarding the broader literature, TokenLearner and DynamicViT address token reduction in vision transformers, where semantically meaningful patches already exist. In sensing signals, the fundamental challenge is that there are no natural semantic units, and the tokenization itself must be constructed from continuous waveforms. Dywave addresses this tokenization gap by grounding segmentation in physics-informed wavelet representations, which is complementary to but distinct from vision-based token selection. Moreover, we position Dywave as the tokenizer before the backbone encoder, modularizing the token reduction and the encoding process. We will expand the related work section to discuss these connections in the revised manuscript.
>
> ---
> **W2. Semantic alignment is not objectively demonstrated beyond visual examples.**
>
> We appreciate the suggestion for more rigorous validation. However, fine-grained boundary annotations are generally unavailable or ill-defined for IoT sensing signals. Unlike images or text, where semantic units are visually or linguistically explicit, raw sensor waveforms are not directly interpretable to human annotators, and annotations derived from other modalities (e.g., video) may not accurately reflect what the sensors capture. This challenge is precisely the *tokenization gap* that Dywave aims to address. With these constraints, we evaluate semantic alignment through its *downstream impact*. Dywave produces substantially more compact token representations while consistently improving performance and cross-domain generalization. These gains indicate that the learned segmentation captures task-relevant structure, as misaligned boundaries would degrade performance under compression. Notably, this is achieved **without any boundary supervision**, suggesting that Dywave discovers meaningful structure directly from the data.
>
> Developing more direct evaluation protocols for semantic alignment in sensing signals is an important direction for future work, and we will include this discussion in the revision.
>
> ---
> **W3 + Q2. Multimodal anchor selection — per-modality or joint?**
>
> We follow the standard approach in multi-modal sensing where each modality is processed by a separate encoder perform late fusion. Specifically, Dywave produces modality-specific tokens that are fed into separate backbone encoders. The final predictions are aggregated across modalities. This per-modality design is due to that different modalities exhibit different temporal dynamics for the same physical event, and forcing joint anchor selection could compromise the adaptive tokenization for individual modalities. This allows Dywave to respect each signal's intrinsic temporal structure while downstream fusion captures cross-modal interactions. We acknowledge that explicit multimodal tokenization is a key future work in the Discussion section and will clarify the multimodal experiment setup in the revised manuscript.
>
> ---
> **Q1. Overhead**
>
> Due to space limit, we kindly refer the reviewer to item 1 of our rebuttal response to Reviewer BNSG, where we provide a detailed wall-clock and memory comparison.
>
> ---
> **W4. Format.**: We have fixed this formatting issue and will double-check the revised manuscript. Thank you for catching this.

---

> > ### Author Rebuttal · Reviewer_spNB · 2026-04-05
> >
> > The authors rebuttal mostly addressed my concerns.
> > Therefore, I increased the score to 4.
> > However, I still see potential for a more detailed evaluation to better underline the claims.

---

> > > ### Author Response · Authors · 2026-04-06
> > >
> > > We sincerely thank the reviewer for the thoughtful feedback and for recognizing the improvements in our rebuttal. We appreciate the increased score and are encouraged that our clarifications have addressed the primary concerns, and we are happy to engage in further discussions if helpful.

---

### Official Review · Reviewer_GmA7 · 2026-03-12

**Soundness:** 3
**Presentation:** 3
**Significance:** 2
**Originality:** 2
**Overall Recommendation:** 4
**Confidence:** 4

**Summary:**

This paper introduces Dywave, a novel dynamic tokenization framework designed to handle the complexities of heterogeneous IoT sensing signals. The authors argue that traditional uniform, fixed-length patching methods are ill-suited for continuous time-series data, which often lacks the natural discrete units found in text or images. To address this, Dywave utilizes the Maximal Overlap Discrete Wavelet Transform (MODWT) to generate hierarchical time-frequency representations. These representations are then processed by detail and context encoders to produce hierarchical embeddings. The framework estimates temporal event saliency by analyzing changes in embedding similarity between adjacent timesteps, allowing for the selection of anchors at potential event transitions. Finally, a dynamic temporal fusion mechanism aggregates signal segments around these anchors using saliency-weighted pooling, resulting in a significantly compressed token sequence. The method was evaluated across five IoT sensing datasets, demonstrating improved classification accuracy and a reduction in token counts by up to 75% compared to standard Transformer and Mamba2 baselines. The study also indicates enhanced cross-domain generalization and robustness to variations in sequence length.

**Compliance With Llm Reviewing Policy:**

Affirmed.

**Final Justification:**

After reviewing the rebuttal and discussion, my concerns regarding experimental validation and design justification have been largely resolved. The authors provided helpful clarifications on reconstruction loss and strengthened the baseline comparisons, which demonstrates the method's robustness.

While deeper sensitivity analyses on the saliency and wavelet tokenization would further strengthen the work, the core contribution, an efficient, structured representation learning framework, is technically sound and meaningful. The rebuttal has increased my confidence in the paper's value, and I now recommend weak acceptance.

**Key Questions For Authors:**

- Can you provide a technical justification for why cosine similarity between adjacent embeddings is a superior proxy for event boundaries compared to classical change-point detection methods?
- What is the sensitivity of the model to the anchor budget parameter $\tau$, and how does performance degrade if the number of anchors is incorrectly specified for a given task?
- What is the specific computational overhead introduced by the MODWT and hierarchical embedding layers when compared to simple patching?
- How does Dywave perform on time-series forecasting tasks, where effective tokenization is a critical component of state-of-the-art architectures?
- Is it possible to train the tokenization module in an end-to-end fashion without the explicit inclusion of the wavelet decomposition?
- To what extent does the reconstruction loss conflict with the objective of achieving a highly compressed and efficient token representation?

**Limitations:**

- The paper's discussion on societal impact and privacy is relatively brief. While the authors acknowledge that high-fidelity sensing can lead to surveillance risks and monitoring without consent, a more comprehensive analysis of how tokenization affects data privacy would be beneficial.
- The study is primarily focused on classification, leaving the generalizability of Dywave to other common time-series tasks, such as anomaly detection or forecasting, largely unaddressed.
- The reliance on specific signal processing techniques like MODWT may limit the framework's accessibility for researchers without a background in traditional signal analysis.

**Strengths And Weaknesses:**

Strengths
- The research addresses a significant and well-defined gap in sensing signal processing by identifying the limitations of uniform windowing in non-stationary IoT data.
- The framework is designed to be modular and architecture-agnostic, ensuring compatibility with various downstream models such as Transformers and state-space models like Mamba2.
- The integration of MODWT is technically sound, as it leverages established signal-processing principles to capture multi-resolution dynamics in non-stationary signals.
- The use of saliency-based anchor selection provides an effective mechanism for adaptive token density, allowing the model to focus computational resources on regions where signal dynamics change most rapidly.
- By compressing redundant segments, the approach offers a reasonable strategy for improving the efficiency of large-scale sensing tasks.

Weaknesses
- The distinction between Dywave and existing adaptive tokenization or dynamic pooling methods is not sufficiently articulated. The current novelty appears limited to the specific combination of wavelet features and saliency-based anchors, which may be viewed as an incremental advancement.
- There is a lack of rigorous theoretical or statistical justification for the chosen saliency formulation. The paper does not explain why cosine similarity changes are an optimal proxy for event boundaries or how this measure compares to established metrics like gradient magnitude or spectral energy changes.
- The anchor selection process relies heavily on heuristic design parameters. This contradicts the authors' claim that the method eliminates the need for hyperparameter tuning, as the algorithm remains dependent on several manual configurations.
- The evaluation lacks comparisons against state-of-the-art time-series baselines, making it difficult to determine if the reported performance gains stem from the tokenization strategy itself or simply from superior feature preprocessing.
- The robustness of the saliency estimation is not thoroughly explored, particularly regarding its sensitivity to noise or sensor artifacts that are common in real-world IoT deployments.
- The relationship between the reconstruction loss and the compression goals is unclear, as an emphasis on reconstruction may inadvertently encourage the retention of redundant information that the tokenization aims to remove.

---

> ### Author Rebuttal · Authors · 2026-03-29
>
> We thank Reviewer GmA7 for the detailed and insightful review. We address each point below.
>
> ---
> **W1+W4: Distinction from existing methods & additional baselines**
>
> Dywave's novelty lies in redefining tokenization for sensing as semantic-dependent, rather than a preprocessing or heuristic step. Existing approaches treat tokenization as independent, operating in raw signal space or using hand-crafted criteria. In contrast, Dywave's tokens emerge from learned representations aligned with intrinsic event structure. Unlike dynamic pooling methods, Dywave addresses the tokenization gap by grounding segmentation in physics-informed representations. We added baselines (LightGTS and Ruptures) to demonstrate that Dywave is distinct from and superior to both sample-level adaptive patching and signal-level change-point detection. Due to space limit, we kindly refer the reviewer to our response to **Reviewer spNB W1+Q3** for a detailed comparison. Empirically, dynamic content-adaptive tokenization consistently outperforms fixed patching across diverse IoT tasks, backbones, and settings (short/long context, cross-domain, multi-modal), establishing the tokenization gap as an addressable bottleneck.
>
> ---
> **W2+Q1: Cosine similarity**
>
> Cosine similarity between adjacent embeddings detects semantic transitions in token representations, rather than low-level signal discontinuities. Classical change-point methods operate on raw signals and rely on statistical variations, which may not correspond to meaningful events in heterogeneous sensing data. Dywave instead constructs hierarchical embeddings integrating multi-scale temporal and contextual information, then identifies transitions in this semantically structured space. Cosine similarity captures shifts in the underlying semantic state, detecting event boundaries reflecting activity changes rather than superficial fluctuations. It also incurs lower computational overhead, requiring only a single pass over the sequence without iterative optimization. Our comparison with Ruptures confirms this: Ruptures achieves substantially lower accuracy while incurring higher cost due to per-sample optimization, demonstrating that representation-driven saliency is both more aligned and more efficient.
>
> ---
> **W3+Q2: Hyperparameter sensitivity.**
>
> Due to space limit, we kindly refer the reviewer to our response to **Reviewer BNsG Q2 and Q5** for a detailed sensitivity analysis. PatchTST's hyperparameters directly determine tokenization structure and require extensive grid search. In contrast, Dywave's anchor budget and λ\_rec serve as regularizers while actual boundaries are determined adaptively, and performance is generally robust to their choice.
>
> ---
> **W5: Robustness to noise**
>
> While we do not include a dedicated noise-injection study, cross-domain generalization experiments provide indirect evidence, as signal characteristics differ significantly between domains. Dywave maintains strong performance with substantially fewer tokens under these shifts. Methodologically, saliency estimation is inherently robust: it operates on hierarchical embeddings integrating multi-scale context; cosine similarity is invariant to magnitude scaling; and anchor selection suppresses isolated spurious peaks. We will discuss noise robustness in the revised paper.
>
> ---
> **W6 + Q6. Reconstruction loss and compression goals.**
>
> Reconstruction and compression serve complementary roles. Reconstruction regularizes fused tokens to preserve multi-scale signal structure, ensuring compression does not discard semantically important information. Without it (Dywave-w/oRecon), performance degrades noticeably, confirming that reconstruction encourages semantically coherent compression. The decoder is discarded at inference, adding no overhead. Our ablation over λ\_rec (**Reviewer BNsG Q5**) shows stable performance.
>
> ---
> **Q3: Computational overhead.**
>
> Due to space limit, we kindly refer the reviewer to item 1 of our response to **Reviewer BNsG Q1** for wall-clock and memory comparisons, demonstrating that preprocessing cost is amortized by downstream savings and the advantage scales with sequence length.
>
> ---
> **Q4: Time-series forecasting.**： Dywave is designed for event-aligned representation learning in heterogeneous IoT sensing, where the primary task is classification. We position it as a tokenization framework for IoT sensing classification rather than a universal time-series method. Our evaluation across five datasets, two backbones, and multiple settings demonstrates broad applicability within this domain. Extending dynamic tokenization to forecasting is an interesting future direction we will discuss it in the revised manuscript.
>
> ---
> **Q5: Training w/o wavelet.** Yes. We evaluate this in the ablation (Dywave-w/oWave). Removing wavelet-based hierarchical embedding leads to consistently worse performance and more tokens, indicating the model remains trainable end-to-end but loses both efficiency and accuracy.

---

> > ### Author Rebuttal · Reviewer_GmA7 · 2026-04-04
> >
> > I thank the authors for their detailed response and the additional empirical justifications provided. The authors have satisfactorily addressed concerns regarding the framework's positioning against baselines, the regularizing role of reconstruction loss, the necessity of wavelet-based embeddings, and the current focus on IoT classification tasks.
> >
> > Partially Resolved Concerns & Follow-ups
> > - While the intuition for cosine similarity is helpful, the justification requires quantitative comparisons against alternative saliency measures like gradient magnitude or spectral energy.
> > - Sensitivity to the anchor budget $\tau$ should be evidenced through explicit plots to demonstrate robustness against over- or under-estimation of event counts.
> > - Direct noise injection or sensor perturbation experiments are needed to validate saliency estimation robustness under realistic IoT noise and artifacts.
> > - The specific computational overhead of MODWT and hierarchical embeddings should be reported to identify the sequence lengths where Dywave yields net benefits.
> >
> > Remaining Questions
> > - Could the authors provide controlled ablations to isolate the primary performance driver among wavelet representation, saliency-based tokenization, and fusion?
> > - How does Dywave differ conceptually from recent adaptive tokenization methods beyond the use of wavelet features?
> > - Can the trade-off between reconstruction fidelity and compression ratio be quantified, specifically regarding whether higher $\lambda_{rec}$ values retain redundant tokens?
> > - Does the method generalize to domains with significantly different temporal dynamics or tasks like anomaly detection?

---

> > > ### Author Response · Authors · 2026-04-05
> > >
> > > We sincerely appreciate your constructive response. To address your concerns, we conducted additional experiments and clarifications below.
> > >
> > > ---
> > > **C1 Alternative saliency**: We added a spectral energy similarity measure as an alternative saliency criterion and compare it against Dywave's cosine-based saliency across all four datasets.
> > > ||Seismic|||Audio|||RWHAR|||PAMAP2|||
> > > |-|-|-|-|-|-|-|-|-|-|-|-|-|
> > > |Method|Acc|F1|#Tok|Acc|F1|#Tok|Acc|F1|#Tok|Acc|F1|#Tok|
> > > |Spectral|76.83|76.12|25.12|85.33|85.09|10.18|82.17|82.30|9.51|72.00|71.79|4.74|
> > > |Cosine|79.17|79.22|16.90|90.02|90.01|50.34|90.94|89.32|29.61|79.77|79.05|3.76|
> > >
> > > Across all datasets, semantic similarity consistently yields higher performance, confirming that semantic transitions as more informative for event-aligned tokenization. The spectral measure yields a strong ablation baseline and occasionally produces lower #tok. We appreciate the reviewer's insightful suggestion, and will include further discussion in the revised manuscript to consider a hybrid criterion combining these as a promising future extension.
> > >
> > > ---
> > > **C2 Sensitivity to anchor budget**: We thank the reviewer for this suggestion. While ground-truth event boundaries are unavailable, we find the anchor budget does not scale linearly since it only defines an upper bound. Anchor selection is data-adaptive, and budget acts as safe upper bound. Due to space limit, we will include plots of performance and #tok vs. budget in the revised manuscript to further visualize this.
> > >
> > > ---
> > > **C3 Robustness under noise and perturbations**: We conducted controlled noise-injection experiments on RWHAR by adding Gaussian noise of varying magnitude at test time.
> > > |Noise|PatchTST (#tok=109) Acc|F1|Dywave Acc|F1|#Tok|
> > > |-|-|-|-|-|-|
> > > |0|73.40|74.29|85.17|82.25|34.65|
> > > |0.05|71.96|68.17|85.46|82.48|38.91|
> > > |0.10|65.32|60.04|82.29|79.59|41.55|
> > > |0.15|55.91|48.51|76.00|75.47|43.97|
> > > |0.20|43.91|35.53|69.65|68.18|45.99|
> > >
> > > PatchTST degrades sharply as noise increases, while Dywave remains substantially more robust across all noise levels. Notably, Dywave adaptively increases its #tok under noise (from 34.65 to 45.99), reflecting dynamic allocation of representational capacity when signal uncertainty grows. In the revised manuscript, we will also explicitly discuss noise robustness.
> > >
> > > ---
> > > **C4 Component overhead and efficiency**: While Dywave introduces preprocessing overhead relative to heuristic patching, this cost is amortized by substantial reductions in backbone computation, and the advantage grows with context length.
> > >
> > > ||Ego4D(30s)|PAMAP2(5s)|RWHAR(4.5s)|MOD(2s)|
> > > |-|-|-|-|-|
> > > |Dywave|||||
> > > |DWT|0.0041|0.0018|0.0023|0.0014|
> > > |Encode|0.1125|0.0666|0.0414|0.0180|
> > > |Saliency|0.0512|0.0098|0.0126|0.0057|
> > > |Merge|0.0199|0.0096|0.0135|0.0087|
> > > |Backbone|0.0640|0.0401|0.0616|0.0705|
> > > |PatchTST|||||
> > > |Patch|0.0021|0.0013|0.0025|0.0012|
> > > |Backbone|0.3977|0.1558|0.2028|0.0832|
> > >
> > > ---
> > > **Q1 Controlled ablations**: We kindly direct the reviewer to the ablations in the main manuscript, which includes w/oWave (removing wavelet hierarchical representations) and w/oFusion (removing fusion); since fusion depends on saliency, we isolate saliency via CNNBound or only applies wavelet hierarhical representations to PatchTST via FixedDWT. We will clarify these methods with better descriptions in the revised manuscript.
> > >
> > > ---
> > > **Q2 Conceptual distinction from adaptive tokenization**: Dywave differs from prior adaptive tokenization in two ways. First, it performs event-aligned tokenization that jointly considers inter- and intra-sequence differences, locating segments that are semantically meaningful rather than merely statistically distinct. Existing methods such as LightGTS focus primarily on inter-sequence variation and overlook how temporal dynamics evolve within a single sequence. Second, Dywave requires no boundary ground truth. Boundaries emerge adaptively from the learned representation and are shaped by the downstream task, yielding tokenization that is both boundary-free and task-aligned.
> > >
> > > ---
> > > **Q3 λ vs. #tok trade-off**: Token count is non-monotonic in λ and does not necessarily increase with higher λ. Reconstruction regularizes the fused representations to preserve multi-scale structure, while token allocation is determined by saliency. Since saliency depends on semantic changes in representation, higher λ does not increase anchor density. Thus, reconstruction can improve fidelity without inflating token count.
> > > |λ|RWHAR #tok|PAMAP2 #tok|
> > > |-|-|-|
> > > |0.1|32.24|2.68|
> > > |0.2|57.24|2.18|
> > > |0.3|56.74|2.30|
> > > |0.5|34.65|2.23|
> > > |0.7|51.13|2.05|
> > > |1.0|35.36|2.06|
> > >
> > > ---
> > > **Q4 Generalization to other domains**: Our evaluation targets classification, but since Dywave is trained jointly with the domain-specific task objective, it can be trained with other downstream objectives. A fully domain-agnostic tokenizer that handles diverse temporal dynamics via self-supervised learning is an important direction we pursue, and we will explicitly discuss it in the revised manuscript.

---

### Official Review · Reviewer_GdfA · 2026-03-13

**Soundness:** 3
**Presentation:** 3
**Significance:** 3
**Originality:** 3
**Overall Recommendation:** 4
**Confidence:** 3

**Summary:**

This paper introduces Dywave, a dynamic, event-aligned tokenization framework for heterogeneous IoT sensing signals. The method leverages MODWT-based hierarchical embeddings (detail via lightweight CNNs, context via an hourglass transformer), computes temporal saliency from adjacent-embedding dissimilarity to select anchors via NMS + TopK, and performs saliency-weighted fusion around anchors to produce compact, variable-length token sequences, with a reconstruction auxiliary loss on wavelet coefficients. Across five datasets and two backbone families (Transformer, Mamba2), Dywave reportedly improves classification accuracy (up to 12%) while reducing token counts by up to 75%, and shows better robustness under domain and sequence-length shifts.

**Compliance With Llm Reviewing Policy:**

Affirmed.

**Key Questions For Authors:**

Questions for Authors
1. How do gradients flow through the anchor selection pipeline (NMS, TopK, nearest-anchor assignment)? Are these steps treated as non-differentiable with stop-gradient, or do you employ a surrogate/soft assignment? Please clarify training dynamics and stability.
2. How is τ (maximum anchor density) selected, and how sensitive is performance to τ and the NMS window size? Can you include a sensitivity/robustness analysis?
3. How are positional encodings handled after dynamic fusion with variable token lengths? Do you use learnable positions relative to anchor indices, and how does this impact transfer across sequence lengths?
4. Can you compare against adaptive segmentation baselines (e.g., Ruptures/BOCPD + segment pooling, learnable temporal pooling/token selection) to strengthen the case for Dywave’s anchor-based approach?
5. How is λrec chosen, and what is the trade-off between reconstruction fidelity and downstream accuracy/efficiency? An ablation over λrec would be helpful.

**Limitations:**

yes

**Strengths And Weaknesses:**

The paper is well-structured, with standardized experiments and figures and good formatting.

---

> ### Author Rebuttal · Authors · 2026-03-29
>
> We thank Reviewer GdfA for the thorough and constructive review. We address each weakness and question below.
>
> ---
> **Q1: Clarification on anchor selection gradients flow:**
>
> These acts as discrete selection and weighting; gradients do not directly propagate through these decisions. Learning occurs through the continuous representations that induce the selection. The task loss backpropagates through the backbone into the fused tokens, which are saliency-weighted aggregations (P_t) of hierarchical embeddings (E_f). Gradients therefore update both E_f and P_t as they directly participate in the weighted fusion. While selection is not differentiable, it is a deterministic function of the learned saliency scores P_t. As E^F and P_t evolve during training, the saliency landscape shifts, leading to different anchor selections across iterations. This forms an implicit (non-gradient) feedback loop, analogous to hard attention or top-k routing where selection is discrete but driven by learned representations.
>
> ---
> **Q2: Anchor Selection Sensitivity Analysis:**
>
> We provide a sensitivity analysis on the maximum anchor budget (determined by τ) below. On RWHAR and PAMAP2 with Mamba2, Dywave maintains stable performance across anchor budgets ranging from 64 to 512.
>
> |Anchor Budget|RWHAR Acc|RWHAR F1|PAMAP2 Acc|PAMAP2 F1|
> |---|---|---|---|---|
> |64|0.8350|0.8090|0.8182|0.8076|
> |128|0.8477|0.8069|0.8068|0.7965|
> |256|0.8517|0.8225|0.8072|0.7980|
> |512|0.8310|0.8067|0.8032|0.7968|
>
> We clarify that the anchor budget does not directly determine exact token boundaries. Boundaries are driven by the learned saliency token representation and act as regularizers preventing over-segmentation. In practice, the final number of tokens is often significantly smaller than the maximum. NMS frequently produces fewer anchors, especially for signals with sparse transitions or long stationary intervals, while TopK serves only as a safeguard against unusually dense saliency peaks. Dywave shows milder sensitivity compared to fixed-patch methods that require extensive grid search over patch size and stride.
>
> ---
> **Q3: Positional encodings clarification:**
>
> We do not apply absolute positional encodings after dynamic fusion. After anchor-based merging, tokens correspond to variable temporal spans, making index-based positional encodings semantically inconsistent across samples. Temporal information is instead preserved through (i) strict ordering of anchors, (ii) temporal context encoded in pre-fusion hierarchical representations (e.g., MODWT and context encoder), and (iii) the sequence modeling backbone. For Mamba2, temporal order is captured via sequential recurrence; for Transformers, ordering is preserved through the structured input sequence without relying on absolute positional indices. This design mitigates length-dependent positional bias and improves robustness to varying sequence lengths. Token counts scale with signal complexity rather than duration, enabling generalization without positional extrapolation.
>
> ---
> **Q4: Additional segmentation baselines:**
>
> We have added comparisons with LightGTS (sequence-adaptive) and Ruptures-based segmentation.
>
> |Backbone|Method|RWHAR Acc|RWHAR F1|# Tokens|PAMAP2 Acc|PAMAP2 F1|# Tokens|
> |---|---|---|---|---|---|---|---|
> |Transformer|LightGTS|0.8586|0.8513|**14.66**|0.6830|0.6621|10.71|
> ||Ruptures|0.7761|0.7302|57.60|0.6877|0.6604|37.95|
> ||Dywave|**0.9094**|**0.8932**|29.61|**0.7977**|**0.7905**|**3.76**|
> |Mamba2|LightGTS|0.7917|0.7409|**14.66**|0.7003|0.6977|10.71|
> ||Ruptures|0.7998|0.7665|57.60|0.6629|0.6341|37.85|
> ||Dywave|**0.8517**|**0.8225**|34.65|**0.8072**|**0.7980**|**2.23**|
>
> LightGTS learns a per-sequence patch size but applies uniform windows within each sequence, overlooking intra-sequence heterogeneity where different regions require different granularity. While it uses fewer tokens on RWHAR, Dywave achieves higher accuracy, indicating these tokens encode meaningful semantics missed by uniform patching. Ruptures relies on statistical change-point detection, which is computationally slow and not task-aware. Dywave outperforms both methods while maintaining practical efficiency. These results will be included in the revision.
>
> ---
> **Q5: Reconstruction loss ablation:**
>
> We provide an ablation over λ_rec below. Performance remains stable across 0.1 to 1.0.
>
> |λ_rec|RWHAR Acc|RWHAR F1|PAMAP2 Acc|PAMAP2 F1|
> |-|-|-|-|-|
> |0.1|0.8506 |0.8259|0.7492|0.7409|
> |0.2|0.8396 |0.8117|0.7764|0.7725|
> |0.3|0.8327 |0.8037|0.7950|0.7907|
> |0.5|0.8517 |0.8225|0.8072|0.7980|
> |0.7|0.8500 |0.8046|0.8091|0.8010|
> |1.0|0.8465 |0.8191|0.8064|0.8038|
>
> The reconstruction loss acts as a regularizer encouraging fused tokens to preserve multi-scale structure rather than maximizing reconstruction fidelity. The decoder is discarded at inference, adding no overhead. The ablation (Dywave-w/oRecon in Table 4) shows that removing reconstruction degrades performance, confirming its role in maintaining representation quality.

---

> > ### Author Rebuttal · Reviewer_GdfA · 2026-04-02
> >
> > Thank you for your detailed rebuttal and clarifications.

---

### Decision · Program_Chairs · 2026-04-30

**Decision:**

Accept (regular)

**Comment:**

This work proposes a dynamic tokenization framework for heterogeneous IoT sensing signals that replaces fixed-size uniform patching with event-aligned, variable-length tokens sequences.

All four reviewers gave weak accepts, with two marking concerns fully resolved (GdfA, spNB), one partially resolved but mostly satisfied with the rebuttal (GmA7), and one maintaining their score while acknowledging the clarifications were helpful (BNsG).

The reviewers highlighted the well-motivated problem formulation and methodology and the thorough evaluation.

The primary concerns centered on computational overhead of preprocessing, hyperparameter sensitivity, comparisons with adaptive segmentation baselines, and noise robustness. The authors addressed these substantively during rebuttal: they provided wall-clock timing and memory comparisons, demonstrated stable performance across anchor budgets, added comparisons against LightGTS and Ruptures (outperforming both), and ran noise-injection experiments showing that their proposed approach is reasonably robust.

I recommend acceptance. The paper identifies principled solution to a reasonable, underexplored problem, and has performed substantial analysis to validate their approach.